# MoGen: Detailed Neuronal Morphology Generation via Point Cloud Flow Matching

**Franz Rieger**[1,2]*, **Jan-Matthis Lueckmann**[1], **Viren Jain**[1] **& Michal Januszewski**[1]
[1]Google Research
[2]Max Planck Institute for Biological Intelligence

riegerfr@bi.mpg.de, {janmatthis,viren,mjanusz}@google.com

## Abstract

Biological neurons come in many shapes. High-fidelity generative modeling of their varied morphologies is challenging yet underexplored in neuroscience, and crucial for the subfield of connectomics. We introduce MoGen (Neuronal Morphology Generation), a flow matching model to generate high-resolution 3D point clouds of mouse cortex axon and dendrite fragments. This is enabled by an adaptation that injects local geometric context into a scalable latent transformer backbone, allowing for the generation of high-fidelity, realistic samples. To assess MoGen's generation quality, we propose a dedicated evaluation suite with interpretable geometric and topological features tailored to neuronal structures that we validate in a user study. MoGen's practical utility is showcased through controllable generation for visualization via smooth interpolation and a direct downstream application: we augment the training set of a shape plausibility classifier from a production connectomics neuron reconstruction pipeline with millions of generated samples, thereby improving classifier accuracy and reducing the number of remaining split and merge errors by 4.4%. We estimate this can reduce manual proofreading labor by over 157 person-years for reconstruction of a full mouse brain.

## 1 Introduction

Modeling the intricate three-dimensional shapes of neurons is a fundamental challenge in neuroscience. Neuron morphologies exhibit tremendous diversity in size, thickness, and branching patterns, as well as compartment-specific properties like dendritic spine density. Variations exist not only between neurons of different species but also within a single brain and even within a single cell type (Ramón y Cajal, 1894; Ascoli, 2002). While generative modeling has revolutionized fields like natural language processing (Brown et al., 2020), image generation (Rombach et al., 2022), and protein structure prediction (Abramson et al., 2024), its application to neuronal morphology remains nascent. Previous data-driven approaches have focused on simplified representations like sparse skeletons (Laturnus & Berens, 2021; Yang et al., 2024) or coarse occupancy maps (Hansel et al., 2024), failing to capture the detailed neuronal surface details crucial for many applications.

This gap is particularly acute in connectomics, the field dedicated to acquiring and analyzing brain wiring diagrams from large, nanometer-resolution microscopy volumes (Sievers et al., 2024; Shapson-Coe et al., 2024; The MICrONS Consortium, 2025; Tavakoli et al., 2025). A core part of building these wiring diagrams is neuron reconstruction. Substantial progress has been made in automating the underlying segmentation process, but residual errors persist. With individual volumes reaching the petabyte scale, manual correction of these segmentation mistakes ("proofreading") has become a primary bottleneck in the field, with estimated costs reaching billions of USD for an upcoming single mouse brain (Jefferis et al., 2023). One of the barriers to improving these automated methods is the limited amount of high-quality training data for the algorithmic components that assess the local plausibility of neuron shapes during reconstruction, a process that includes segmentation and post-processing steps (Januszewski et al., 2025).

---

*Work performed as a Student Researcher at Google Research.

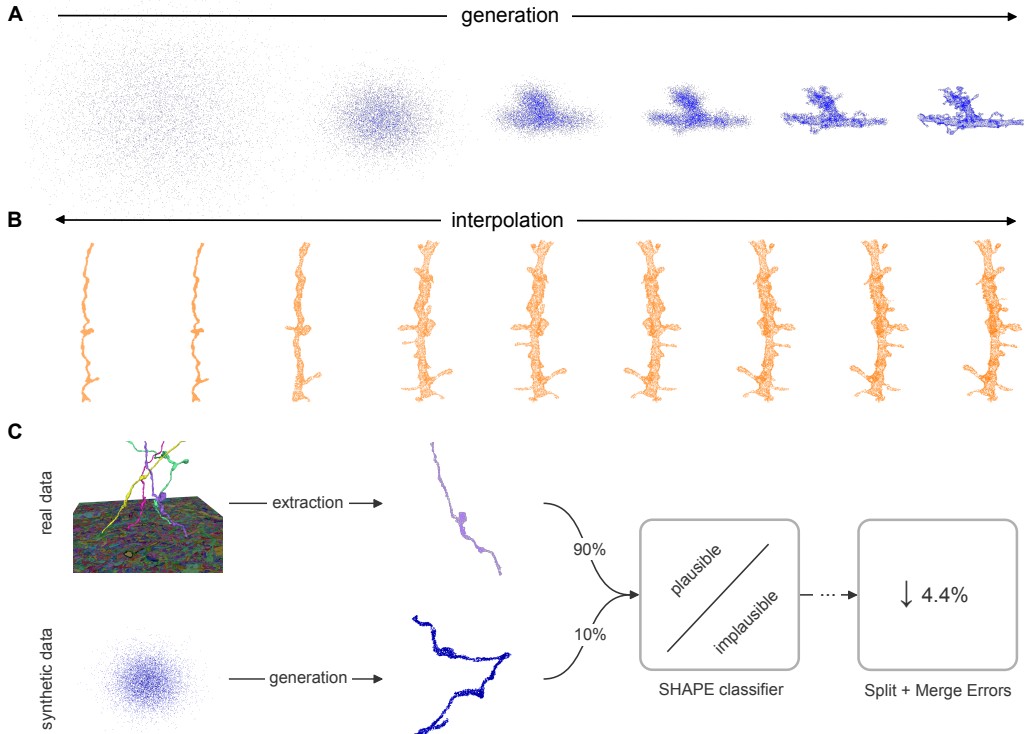

Figure 1: **Overview. (A)** Flow matching on point clouds iteratively transforms Gaussian noise into neuronal shapes. **(B)** MoGen allows for smooth interpolation, here between axon (left) and dendrite (right) fragments. **(C)** Real neuron fragments from an electron microscopy volume are extracted as point clouds. Synthetic examples augment the training data for a downstream shape plausibility classifier which must distinguish between plausible and implausible neuronal morphologies. Co-training improves the classifier, leading to fewer errors in the final neuron reconstruction, which decreases the amount of manual proofreading needed.

This paper directly addresses this data scarcity problem. Neurons have intricate, topologically tree-like structures that extend through a large volume while occupying very little space themselves. For instance, while a neuronal cell body might only be a few tens of microns in diameter, its axon can reach locations that lie many millimeters away (Ascoli, 2002). This geometry makes dense, voxel-based 3D generative approaches computationally- and memory-intensive for high-resolution modeling, motivating the use of more efficient representations such as point clouds. We introduce a generative model, MoGen, capable of producing high-resolution, biologically plausible 3D neuron fragment morphologies. MoGen models neuron morphology as point clouds and employs flow matching for generation (Lipman et al., 2023). To our knowledge, this is the first work to generate high-resolution neuronal morphologies and demonstrate their utility in a critical downstream connectomics task within a production-scale neuron reconstruction pipeline, PATHFINDER (Januszewski et al., 2025). Specifically, we show that by co-training with synthetic data, a strategy that has proven effective in other domains (Azizi et al., 2023), it is possible to improve the accuracy of the SHAPE plausibility classifier used in the PATHFINDER system, leading to a reduction of the number of errors in the final automated reconstruction and directly alleviating the prohibitive costs of manual proofreading (Figure 1).

Our contributions are fourfold:

- **MoGen: a flow matching model for neuronal point clouds:** We adapt a latent transformer backbone to incorporate local geometric context and use flow matching with a custom schedule. This enables generation of high-fidelity morphologies.

- **Dedicated neuron evaluation suite:** We introduce an expressive set of interpretable metrics more topologically tailored to neurites than standard distance metrics used in the point cloud literature. We show in a user study that they correlate well with visual quality.

- **Controllable generation:** We demonstrate that MoGen enables smooth interpolation between morphologies with control over different structural aspects (e.g. branchiness or extent) and diversity of the generated samples through tunable faithfulness to the conditioning.

- **Direct impact on a real-world connectomics pipeline:** We improve a state-of-the-art neuron shape plausibility classifier by co-training with millions of MoGen-generated samples, reducing the number of remaining segmentation errors by 4.4%. At the scale of a complete mouse brain, this corresponds to an estimated savings of about 157 person-years of manual proofreading work — several times more than the total proofreading budget that was used for recently mapped insect brains (Scheffer et al., 2020; Dorkenwald et al., 2022).

## 2 RELATED WORK

### 2.1 AUTOMATED RECONSTRUCTION FOR CONNECTOMICS

Modern connectomics pipelines (Lee et al., 2017; Januszewski et al., 2018; Sheridan et al., 2023; Chen et al., 2025) automate the segmentation of neurons but are prone to errors like incorrect splitting or merging of distinct neurites (Jefferis et al., 2023). To mitigate these errors, often stemming from a limited field of view of the segmentation models, some downstream methods leverage larger-scale shape reasoning (Troidl et al., 2024; Januszewski et al., 2025), or neurite directionality (Schmidt et al., 2024) for automated error correction. In particular, the PATHFINDER pipeline (Januszewski et al., 2025) uses a classifier, called SHAPE, to assess the biological plausibility of agglomerated segments by operating directly on 3D point clouds sampled from the neurite fragment meshes. This process starts with an over-segmented volume, where single neurons are initially broken into multiple smaller segments. This volume is then iteratively agglomerated (merged); the classifier is used to decide which merges are plausible. This classifier is trained to distinguish plausible "positive" examples from implausible "negative" ones created by artificially merging two distinct neurite fragments. As shown by Januszewski et al. (2025, Fig. S2.D), the performance of these critical components is limited by the amount of available high-quality, proofread training examples — a bottleneck MoGen directly addresses.

### 2.2 GENERATIVE MODELS FOR NEURONAL MORPHOLOGY

Data-driven generative modeling of neuron morphology is an emerging field (Farhoodi & Kording, 2018; Tang et al., 2020; Kanari et al., 2022). Much prior work has relied on procedural, rule-based generation (McCormick & Mulchandani, 1994) or biophysical simulators (Breitwieser et al., 2022), without taking full advantage of learning from data. Recent deep learning approaches have focused on simplified representations. Many global neuron topology modeling approaches focus on neuron skeletons (Cuntz et al., 2010; Zhu et al., 2025). Tang et al. (2020) generate images of neuron skeletons to aid segmentation. MorphGrower (Yang et al., 2024) and MorphVAE (Laturnus & Berens, 2021) generate sparse skeletons, capturing branching topology but not detailed 3D morphology. PointNeuron represents neuron skeletons as connected points with a radius to coarsely model membrane surfaces (Zhao et al., 2023). Similarly, Wiesner et al. (2022) also model dynamic cell surfaces with implicit neural representations and MorphOcc generates 3D occupancy maps, again missing fine detail during generation (Hansel et al., 2024). A key distinction is that these methods typically generate full neurites, e.g., a whole dendritic branch, whereas we focus on much higher-resolution fragments with detailed local (surface) shape, as shown in Figure 2. Accurate surface modeling is vital for simulations (Eyal et al., 2018) and reconstruction tasks where high-frequency details (e.g., spine heads, caliber changes) determine connectivity. Our work is, to our knowledge, the first to generate high-resolution point clouds of detailed neuron fragments and to demonstrate its utility in neuron reconstruction from volume electron microscopy imagery, a downstream scientific application.

### 2.3 GENERATIVE MODELS FOR 3D POINT CLOUDS

We choose point clouds as our representation, as they efficiently encode the sparse, high-frequency spatial information of neuronal membranes without the cubic memory cost of voxel grids. The efficacy of point-based learning for detailed neuronal geometry processing has been previously demon-

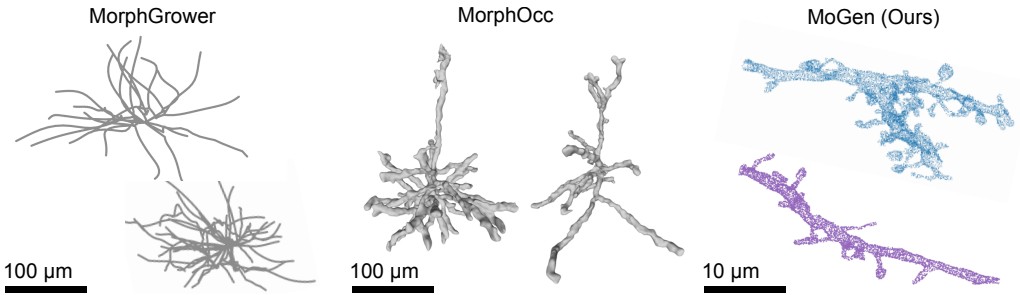

Figure 2: **Comparison with prior work.** Previously proposed generative models of neuron morphology such as MorphGrower (adapted from Yang et al., 2024) and MorphOcc (adapted from Hansel et al., 2024), generate coarse representations (skeletons and occupancy maps) and do not capture fine geometry. In contrast, neurite fragments generated with MoGen show superior details such as dendritic spines. Scale bars are approximate.

strated in reconstruction tasks (Zhao et al., 2023); we extend this paradigm to generative modeling. While generative approaches like generative adversarial networks, variational autoencoders, and diffusion models have been adapted for point clouds (Li et al., 2018; Anvekar et al., 2022; Luo & Hu, 2021), recent flow matching approaches offer simpler training objectives and efficient inference for high quality generation (Lipman et al., 2023; Yushi et al., 2025). However, many general-purpose models rely on architectures with quadratic computational complexity and do not scale well to the large point counts (here, 8192) required for detailed neuronal features. This makes direct quantitative comparison with many existing baselines computationally infeasible at our target resolution. The PointInfinity backbone, initially introduced for diffusion, is a notable exception and considered state of the art for large point clouds, achieving linear scaling by using cross-attention between point tokens and a fixed-size set of latent tokens (Huang et al., 2024). MoGen adapts the PointInfinity architecture with modifications (as outlined below) that improve generation quality of high-fidelity neuronal morphologies. For reference, we also include a comparison to a transformer baseline on a downsampled dataset in Appendix F.1.

## 3 METHOD

We generate neuron morphologies as point clouds in a two-stage process: a flow matching model generates the 3D point coordinates, optionally followed by a regression model predicting per-point features (see Appendix A.4).

### 3.1 POINT CLOUD GENERATION VIA CONDITIONAL FLOW MATCHING

We employ flow matching, for which a vector field $\mathbf{v}_\theta$ transports samples from a simple prior distribution $P_0$ (e.g., a standard Gaussian) to a complex real data distribution $P_1$. Generation involves integrating the Ordinary Differential Equation (ODE) $\frac{d\mathbf{x}_t}{dt} = \mathbf{v}_\theta(\mathbf{x}_t, t, \mathbf{c})$ from $t = 0$ to $t = 1$, starting from a noise sample $\mathbf{x}_0 \sim P_0$. Here, $\mathbf{c}$ is an optional conditioning vector. We train the model $\mathbf{v}_\theta$ to predict the constant-velocity vector $\mathbf{v} = \mathbf{x}_1 - \mathbf{x}_0$ for interpolated samples $\mathbf{x}_t = (1 - t)\mathbf{x}_0 + t\mathbf{x}_1$, where $\mathbf{x}_1 \sim P_1$ is a real data sample. This yields a simple mean squared error objective (Lipman et al., 2023),

$$\mathcal{L} = \mathbb{E}_{t, \mathbf{x}_0, \mathbf{x}_1, \mathbf{c}} \left\| \mathbf{v} - \mathbf{v}_\theta(\mathbf{x}_t, t, \mathbf{c}) \right\|^2, \tag{1}$$

relative to which we optimize the model's parameters $\theta$. We sample the timestep $t$ using a modified cosine schedule (see Appendix A.3) and use a midpoint ODE solver for inference.

### 3.2 IMPROVING MORPHOLOGICAL FIDELITY BY APPENDING RELATIVE COORDINATES

Our vector field model $\mathbf{v}_\theta$ is built upon the scalable PointInfinity architecture (Huang et al., 2024). We identified a critical limitation in the architecture for our domain: its global cross-attention mechanism means each point token's contribution to the latent tokens is independent of its geometric neighbors, causing an information bottleneck (see Appendix A.2 for details). This lack of direct

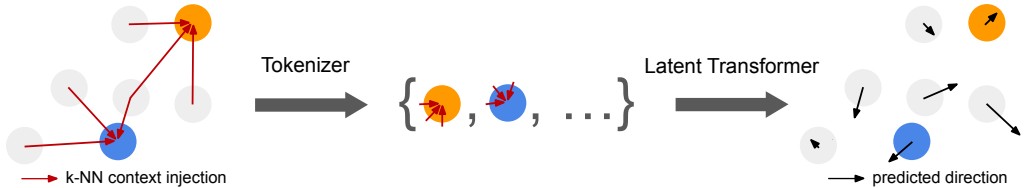

Figure 3: **Architectural adaptation.** MoGen injects local geometric context by finding the k-Nearest Neighbors (k-NN) for each point and appending their relative coordinates as features. This information, along with the point's own coordinates, is processed into tokens that attend to a set of latent tokens in a transformer architecture, enabling the generation of high-fidelity morphology. The model predicts for each point the direction it will move into during integration.

local communication is detrimental, resulting in point clouds with artifacts such as detached (groups of) points. We introduce a simple yet effective modification: for each input point, we find its $k$-Nearest Neighbors (k-NN) and append their relative 3D coordinates as additional features to its token (Figure 3). This well-established principle from the point cloud literature (Qi et al., 2017; Qian et al., 2022) provides a strong *local geometric inductive bias* that proved essential for this architecture to generate high-fidelity point clouds. This modification is computationally lean, adding only a small ($< 15\%$) increase in training time for a substantial gain in quality. This gain is quantifiable both by our metrics and the model loss in the low-noise regime where fine details are formed (see Table 1).

## 3.3 CONTROLLABLE GENERATION

To enable fine-grained control, we train a conditional model where a (projected) vector **c** is concatenated to the latent stream tokens, similar to the conditioning tokens in PointInfinity (Huang et al., 2024). This allows global shape properties to be manipulated, including (Figure 4):

- **Position & Spread (9 features):** The 3D mean and the 6 unique elements of the covariance matrix of the point clouds, computed after mean-centering.
- **Branching Complexity (1 feature):** The number of leaves in a Minimum Spanning Tree (MST) on a 256-point subsample (via farthest point sampling), serving as a proxy for the number of terminal branches.
- **Neurite Type (if present, 1 feature):** An encoding for axon vs. dendrite (e.g., +1 or -1).

To enable unconditioned generation and conditioning on a subset of the vector **c**, we also provide a binary mask indicating which dimensions of **c** are active (details in Appendix B.1). This provides smooth control over the generated shapes, as shown in the interpolation experiments (Figure 5).

## 4 DATA AND EVALUATION

### 4.1 NEURON FRAGMENT DATASET

For our experiments, we use a previously reported (Januszewski et al., 2025) electron microscopy volume of mouse cortex tissue segmented with flood-filling networks (Januszewski et al., 2018). All neurites in this volume have been proofread, and we followed the SHAPE training/validation setup used by Januszewski et al. (2025), with 1,795 axons used for training and 263 for validation. Same as for SHAPE, the volumetric axon reconstructions were meshed with marching cubes (Lorensen & Cline, 1998) and skeletonized with TEASAR (Sato et al., 2000). We then randomly sampled points from the triangular faces of the mesh, restricting the sampling to $10\,\mu\text{m}$ radius spheres centered at the nodes of the skeleton. This center node becomes the origin (0,0,0) of the point cloud, and coordinates are normalized such that 1 unit corresponds to the $10\,\mu\text{m}$ radius. This defines a local coordinate system relative to the fragment center, allowing the model to learn local morphology independent of absolute volume coordinates. Every point cloud was subsequently resampled to a standard size of 8,192 points with farthest point subsampling or random point repetition.

The axons were used to create two datasets of "positive" (plausible) fragments taken from proofread reconstructions and "negative" (implausible), artificially merged examples for training the SHAPE classifier. We also curated a set of 2,200 dendrites, which we split into training (2,000) and validation (200) sets. We added the dendrites to the "positive" axons to get a "mixed" dataset of plausible neurites. This combined dataset provides a challenging and realistic benchmark for generative modeling of cortical cell morphology.

We trained different MoGen models for different purposes: a general model on the mixed set for most visualizations and ablations, and a separate model trained only on negative axon examples (see Figure 14) for the downstream SHAPE co-training task.

## 4.2 Evaluation Protocol: A Custom Metric Suite

Standard metrics for point cloud generation (Huang et al., 2024), like Chamfer Distance (which, for two point clouds, averages the distance from each point in one cloud to its nearest neighbor in the other), are poor indicators of quality for neurons. A neurite can branch in numerous valid ways, and a small, biologically plausible change in the angle of a branch can lead to a large Chamfer Distance, unfairly penalizing a high-quality sample (see Appendix D.1 for examples). Other common metrics, like Fréchet Inception Distance (FID) on rendered images or features from pre-trained networks (Yushi et al., 2025), may suffer from domain shift and do not explicitly capture the specific topological and geometric properties critical for neurons.

We therefore designed a custom evaluation suite that compares the distribution of 10 interpretable features between real and generated sets. Because neuron fragments in our dataset have no canonical orientation and we use rotation augmentations, our evaluation features are designed to be rotation-invariant. We use Maximum Mean Discrepancy (MMD) (Gretton et al., 2012) as it requires fewer assumptions than Fréchet Distance (Jayasumana et al., 2024). The features, which we selected based on empirically observed generation failures in early experiments, are (see Figure 4):

- **Global Shape (4 features):** Distance of the center of mass to the origin; standard deviation of point coordinates along the first, second, and third principal components. These capture overall size and elongation.

- **Local and Global Point Density (4 features):** Mean and standard deviation of the distance to the nearest neighbor (capturing local geometric regularity) and farthest neighbor (capturing compactness).

- **Topology (2 features):** As neurites are tree-like structures: total weight (a proxy for total path length) and longest edge (sensitive to fragmentation) of a Minimum Spanning Tree (MST) on all points.

Crucially, we validate that our MMD score (see Appendix D.2 for details) is a meaningful proxy for generation quality. In early experiments we found that the MMD score correlates well with the model's training loss, indicating that the metric captures model improvement. Furthermore, the features themselves are biologically salient: a UMAP (McInnes et al., 2018) projection of our 10D feature vectors for real data (see Figure 9 in the Appendix) shows a clear separation between axon and dendrite morphologies, with a simple linear classifier achieving 99.3% accuracy. This confirms that our features capture meaningful morphological differences. Finally, the MMD score also correlates strongly with human perception of quality, as detailed in our user study (Figure 8 in the Appendix).

## 4.3 Downstream Task Setup

We demonstrate the practical utility of MoGen by extending the training set of the SHAPE classifier from the PATHFINDER pipeline (Januszewski et al., 2025) with synthetic examples. We replicate the original training setup (we start with the same segmentation, and evaluate the number of split and merge errors in the final reconstruction after the SHAPE-guided search for an optimal agglomeration) but randomly sample 10% of the negative examples (as this class of erroneous merges is more complex and diverse) from a pool of 126M MoGen-generated point clouds. All generated samples for this data augmentation were unconditional to maximize diversity. We report further details in-

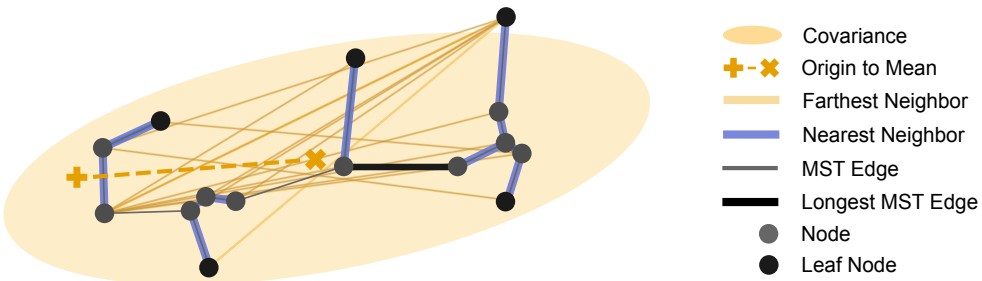

Figure 4: **Evaluation and conditioning feature visualization.** Our interpretable features are divided into three categories. **Global Shape** features (orange) describe the overall size and elongation, including the distance of the center of mass from the origin, the covariance of the point coordinates, and the mean and standard deviation to the farthest neighbors. **Local Point Density** features (blue) measure surface regularity, such as the mean and standard deviation of distances to the nearest neighbors. **Topology** features (black) capture the tree-like structure, using properties of a Minimum Spanning Tree (MST) like its total weight (path length), the length of its longest edge and the number of leaves. Different subsets of the features are used for evaluation and conditioning with formal definitions in Appendix C.

cluding checkpoint selection in Appendix E.1. After training the SHAPE plausibility classifier we use it improve the quality of the axon reconstruction as described in (Januszewski et al., 2025).

## 5    RESULTS AND ANALYSIS

Our experiments validate two key claims: (1) our local context injection and cosine schedule improve generation of high-fidelity morphologies, and (2) our synthetic data provides a tangible, substantial improvement in a real-world scientific application.

### 5.1    GENERATION QUALITY AND CONTROLLABILITY

**Quantitative Comparison.** As shown in Table 1, our full model with k-NN context substantially outperforms the PointInfinity-inspired baseline, achieving a lower MMD score (3.54 vs. 4.27). This indicates a more faithful approximation of the real data distribution. The model also achieves a lower loss in the low-noise regime ($t \in [0.8, 0.9]$), which is responsible for refining fine details. We further evaluated the effect of model width and depth. Reducing either the model's width or depth by half leads to a degradation in performance, confirming that a sufficiently large model is necessary to capture the complexity of the data distribution. Furthermore, a full training run of 1 million steps (compared to 500k for ablations) improved the MMD score of our final model to 3.08. A qualitative comparison of real and generated samples is provided in the appendix in Figure 12 with details in Appendix A.1.

Table 1: **Generation quality evaluation.** Architectural and feature-based ablation results, showing the impact of k-NN context injection, model architecture, and scheduling on generative quality. Each configuration represents a single change from the full MoGen setup. All models were trained for 500k steps.

| Configuration | MMD $\downarrow$ | Loss ($t \in [0.8, 0.9]$) $\downarrow$ |
|---|---|---|
| Ours (Full MoGen) | **3.54** | **0.709** |
| – w/o k-NN (Baseline) | 4.27 | 0.712 |
| – Half Width | 6.97 | 0.711 |
| – Half Depth | 9.94 | 0.711 |
| – w/ Linear schedule | 5.12 | - |

**Controllability.** MoGen allows for smooth, controllable interpolation between shapes by varying the conditioning vectors, as demonstrated in Figure 5 (with details in Appendix B.2). This capability serves as a powerful visualization tool for neuroscientists to explore the morphological space,

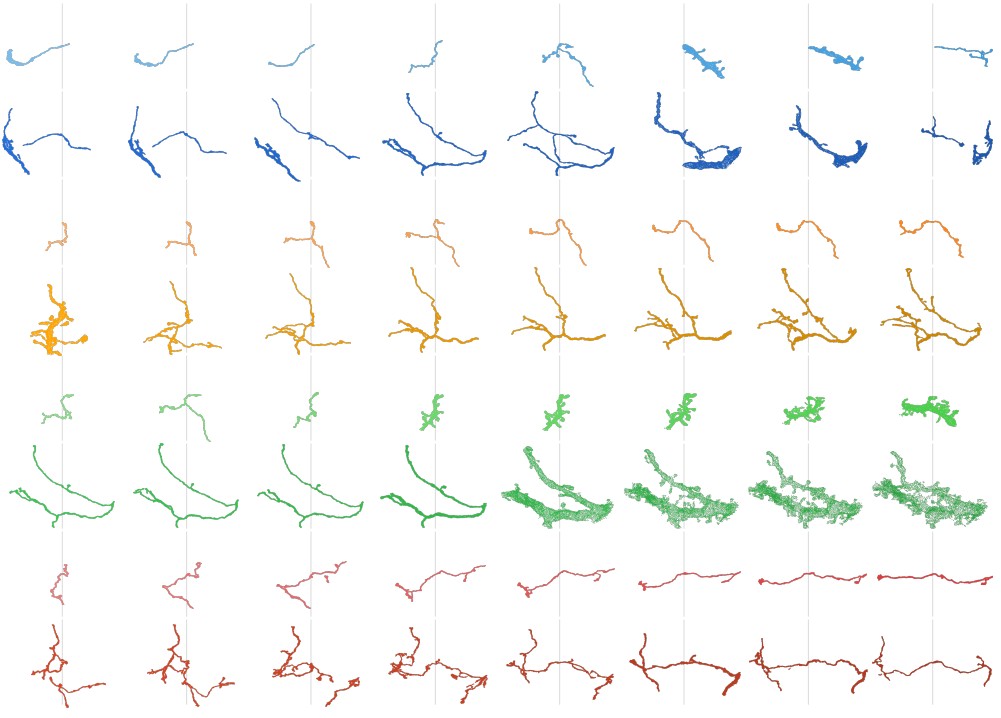

Figure 5: **Controllable interpolation.** Smooth interpolation is achieved by varying individual conditioning features. From top to bottom, pairs of rows show interpolation of: mean x-coordinate (blue), spatial extent (covariance diagonal, orange), branching complexity (#MST leaves, green), and rotation (covariance matrix, red). Note the biologically plausible emergence of fine details, such as dendritic spines, during the interpolation towards more complex dendritic morphologies.

for instance by smoothly interpolating from a simple axon to a complex dendrite with emergent dendritic spines. This demonstrates that the model has learned a structured latent space where high-frequency details are manipulatable. This is biologically relevant as brains contain axon-dendrite hybrids (Goaillard et al., 2020), and it allows neuroscientists to perform counterfactual analysis (e.g., "what would this dendrite look like with higher spine density?") for hypothesis generation. We provide interactive 3D viewers and generation and interpolation animations in the supplementary material for additional qualitative assessment.

As in other domains, using classifier-free guidance (Ho & Salimans, 2022) (which uses the difference between a conditional and an unconditional prediction to guide generation), we also see a trade-off between condition following and diversity in Table 2 (details in Appendix B.3).

Table 2: **Impact of classifier-free guidance scale on generation faithfulness and diversity.** Lower ranks are better for both metrics. The highest guidance scale of 1.0 provides the best faithfulness (MSE rank), while being less diverse (diversity rank).

| Guidance | Faithfulness ↓ | Diversity ↓ |
|---|---|---|
| -1.0 | 5.90 | 3.70 |
| -0.9 | 4.80 | 3.25 |
| -0.8 | 3.75 | 3.20 |
| -0.5 | 3.15 | 3.40 |
| 0.0 | 2.20 | 3.35 |
| 1.0 | 1.20 | 4.10 |

## 5.2 IMPACT ON DOWNSTREAM RECONSTRUCTION PLAUSIBILITY

Having established MoGen's generation quality, we now evaluate its real-world impact.

**Results.** We trained SHAPE models using only real examples as reported before (Januszewski et al., 2025) and with 10% examples synthesized with MoGen (see Appendix E.2 for details). We then used these models in the PATHFINDER pipeline to perform a combinatorial search for an optimal reconstruction of all axons in the volume. The final automated re-

construction is compared to a manually proofread version of the volume, and the total number of remaining split and merge errors is counted. Figure 6 shows that data augmentation with MoGen leads to a decrease in both merge and split errors per millimeter of path length. In the optimal configurations, minimizing the sum of split and merge errors, the error rate is reduced by 4.4% from 0.7947 to 0.7595 errors/mm.

**Economic Impact.** To contextualize this result, a $1\,\mathrm{mm}^3$ volume of mouse cortex is estimated to contain several kilometers of wires (Braitenberg & Schüz, 2013), which corresponds to millions of errors that need to be manually corrected. A whole mouse brain is 500 times larger still. As detailed in Appendix E.4, the improvement due to MoGen thus corresponds to up to 157 person-years of saved manual work in a whole mouse brain project.

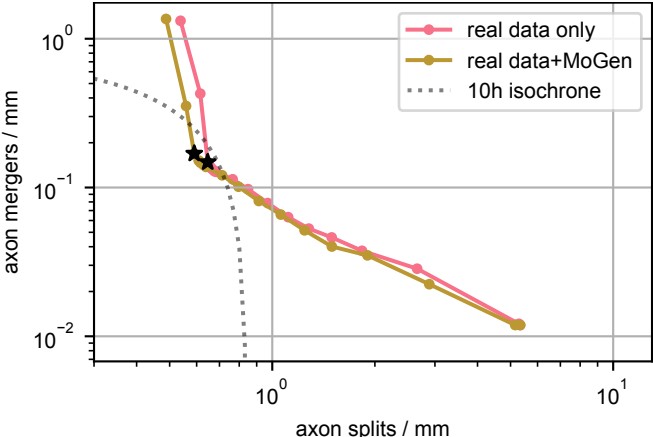

Figure 6: **Error reduction in the final automated reconstruction.** Co-training SHAPE models with MoGen examples simultaneously improves split and merge error rates in PATHFINDER reconstructions and establishes a new Pareto frontier. Stars indicate points that minimize the total number of errors ($0.7947\,\mathrm{mm}^{-1}$ for real data only and $0.7595\,\mathrm{mm}^{-1}$ for real data + MoGen). The curves are obtained by varying a threshold SHAPE score below which an example is considered to have implausible morphology. The dotted line (isochrone) indicates error levels requiring an estimated time of $10\,\mathrm{h}$ to correct.

**Synthetic Data Replacement Fraction.** For the downstream task, we tested different fractions of synthetic data used to replace real examples during training. Table 4 in the Appendix shows that a 10% replacement fraction yields the best performance. This suggests that while synthetic data provides diverse examples that aid generalization, the original data distribution contains unique information essential for achieving peak performance. This phenomenon, where performance degrades if the synthetic data fraction is too high, has also been observed in other domains (Azizi et al., 2023).

## 6 DISCUSSION

This work serves as a case study for generative AI not merely as a tool for creating realistic media, but as an engine for accelerating fundamental scientific research. The ability to generate vast, diverse, and controllable synthetic datasets opens new possibilities for training more robust and accurate models for neuroscience.

**Limitations.** Our primary limitation is the occasional generation of topologically incorrect fragments, such as those with unrealistic cycles or disconnected components (see Appendix Figure 11). The latter can be reduced by post hoc filtering on the longest MST edge. These errors likely stem from the fact that the flow matching loss does not explicitly enforce topological correctness and that there are some topologically disconnected samples in the real data caused by imprecision of the underlying automated reconstruction and image artifacts. Furthermore, our k-NN modification, while improving quality, makes the model resolution-dependent, breaking the invariance of the original PointInfinity architecture (Huang et al., 2024). However, we show in Appendix F.2 that the model generalizes to some degree to higher resolutions not seen during training.

**Synergies and Future Applications.** The potential of this work extends beyond training classifiers. MoGen can be used for visual answers to counterfactual questions such as "what would a neurite with twice the average number of branches look like?" (Figure 5). Furthermore, we demonstrate in Appendix F.5 that MoGen can perform semantic editing of real neuron fragments (e.g., increasing branching on an existing axon) via inverse integration. It could also serve as a visualization tool for future datasets with more features such as cell types per neurite or for counterfactual generation for discovering biologically relevant features similar to Eckstein et al. (2024). Additionally, it could be used for interpreting embedding models trained on other modalities such as gene expression (Gouwens et al., 2019; Yao et al., 2023), where conditioning on a transcriptomic embedding could be translated into a full 3D morphology. If extended to generate multiple neurite fragments in a volume, MoGen could also enable the generation of synthetic segmentation and corresponding EM imagery for augmenting segmentation models (Rieger et al., 2024). A challenge is scaling from fragments of approx. $10\,\mu m$ radius with 8,192 points to complete, multi-millimeter-long neurons. This may require hierarchical approaches that can handle millions of points while maintaining global coherence. Finally, future work could also focus on generating maximally helpful co-training samples instead of just unconditioned ones (Such et al., 2020).

## 7 CONCLUSION

We introduced MoGen, a model for generating detailed, controllable 3D neuron morphologies. We demonstrated its direct, practical value by improving a critical component in a real-world neuron reconstruction pipeline, substantially reducing manual proofreading work, corresponding to over a hundred person-years of saved effort at the scale of a whole mouse brain. Our architectural insight of injecting local geometric context proved beneficial for generating detailed morphology. We validated our approach's fidelity and diversity with a custom metric suite and an extensive user study. This work lays a foundation for a new generation of data-driven tools to tackle the immense scale and complexity of connectomics.

### REPRODUCIBILITY STATEMENT

To ensure the reproducibility of our results, we release the code, together with the pre-trained model weights and the generated neuron fragment data at `https://mogen-release.web.app/`. The appendix provides detailed information on hyperparameters and the training setup.

### LLM USAGE

We used Gemini 2.5/3 Pro to rephrase text at the sentence and paragraph level.

### ACKNOWLEDGMENTS

We thank Peter Li for feedback and Jörgen Kornfeld for support. We gratefully acknowledge the data providers who made this work possible: the Hess lab and the FlyEM team at Janelia Research Campus (for mouse cortex and Drosophila data, respectively), and Jörgen Kornfeld and Winfried Denk (for zebra finch data). F.R. is supported by the Max Planck Society.

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

## A  METHODOLOGY DETAILS

### A.1  FLOW MATCHING DETAILS

Using JAX, we trained our models on 16 NVIDIA A100 GPUs using the Prodigy optimizer (Mishchenko & Defazio, 2024) with a learning rate of 0.5 and a global batch size of 512 for 500k steps for ablations or 1 million steps (approx. 3 weeks) for the full trainings for the visualizations and downstream application. To stabilize training, we used a global gradient clipping norm of 0.1. An exponential moving average (EMA) with a decay of 0.999 was applied to the model parameters, a common practice for stabilizing training in generative models (Ho et al., 2020). We report results for the checkpoint with the best MMD, evaluated every 5,000 steps. Following Januszewski et al. (2025), as data augmentations, we used random rotations and applied a small amount (16 nm stdev.) of Gaussian jitter to the point coordinates. Instead of, as in Huang et al. (2024), training on a lower number of points than generating, we use the same number in both to reduce domain shift. The key hyperparameters for MoGen, yielding 10.7 M trainable parameters, are listed in Table 3.

While for PointInfinity, Huang et al. (2024) describe a latent recursion mechanism for inference, we found that initializing latent tokens with learned values each forward pass was sufficient for high-quality generation.

Table 3: Key model and training hyperparameters.

| Parameter | Value |
|---|---|
| *PointInfinity Architecture* | |
| Point Token Dimension | 128 |
| Latent Token Dimension | 256 |
| Number of Latent Tokens | 256 |
| Number of Stages | 4 |
| Number of Transformer Blocks per Stage | 2 |
| Number of Attention Heads | 8 |
| k for k-NN Context | 16 |
| *Training Parameters* | |
| Optimizer | Prodigy |
| Global Batch Size | 512 |
| Learning Rate | 0.5 |
| Gradient Clip Norm | 0.1 |
| Max Training Steps | 500k (ablations) / 1M (full) |
| EMA Decay | 0.999 |

### A.2  INJECTING A LOCAL GEOMETRIC INDUCTIVE BIAS

While the PointInfinity architecture can model global shape (Huang et al., 2024), its treatment of the input as a set of points can create a bottleneck for generating fine-grained details. The model's "read" cross-attention mechanism aggregates information from all individual point tokens into a fixed-size latent representation. This requires the latent state to compress not only the global structure but also all local surface properties, which can result in a loss of fidelity. To generate high-quality details, the model needs a more direct way to reason about local geometry.

We address this by introducing a strong local inductive bias (Qi et al., 2017). Instead of representing each point $\mathbf{p}_i \in \mathbb{R}^3$ independently, we augment its features with information about its local neighborhood. Let $\mathcal{N}(\mathbf{p}_i) = [\mathbf{p}_{j_1}, \ldots, \mathbf{p}_{j_k}]$ be the (distance sorted) list of the $k$ nearest geometric neighbors of point $\mathbf{p}_i$. We compute the relative coordinates for each neighbor:

$$\Delta\mathbf{p}_{ij_m} = \mathbf{p}_{j_m} - \mathbf{p}_i, \quad \text{for } m = 1, \ldots, k \tag{2}$$

The augmented feature vector $\mathbf{f}_i$ for the point is then the concatenation of its own coordinates and these relative vectors:

$$\mathbf{f}_i = \text{concat}(\mathbf{p}_i, \Delta\mathbf{p}_{ij_1}, \ldots, \Delta\mathbf{p}_{ij_k}) \tag{3}$$

This richer feature vector $\mathbf{f}_i$ is then projected into an input token $\mathbf{t}_i$. This explicit encoding of local patches provides the flow matching model with direct evidence of local structure, such as density, curvature and orientation. It simplifies the learning task, as the model no longer needs to infer these properties solely from the global latent state. This alleviates the information bottleneck, allowing the latent representation to focus on global shape coherence while the augmented input tokens handle local consistency.

### A.3 MODIFIED COSINE SCHEDULE FOR TIMESTEP SAMPLING

For sampling the timestep $t$ during training, we use a modified cosine schedule following Nichol & Dhariwal (2021). It is defined by:

$$C(u) = \begin{cases} 0.5(1 - \cos(\pi u))^2 & \text{if } 0 \leq u < 0.5 \\ 1 - 0.5(1 + \cos(\pi u))^2 & \text{if } 0.5 \leq u \leq 1 \end{cases} \tag{4}$$

We sample $u$ uniformly from $[0, 1]$ and then set the timestep $t = C(u)$. This transforms the uniform distribution to one that concentrates sampling density at the beginning and end of the interval, which we found empirically to balance the learning of global structure (high noise, low $t$) and fine local detail (low noise, high $t$). For inference, we use a midpoint ODE solver with 100 integration steps and this schedule to determine the integration step size.

### A.4 FEATURE PREDICTION MODEL

For downstream tasks that require per-point features originally computed from the source mesh, e.g., pointwise surface normals and local curvature in PATHFINDER's SHAPE models (Januszewski et al., 2025), we train a separate, simple regression model. This model takes a generated (or real) point cloud of 3D coordinates as input and predicts the desired features for each point. It uses the same PointInfinity backbone with k-NN context injection but without time or feature conditioning and is trained with a standard L2 regression loss on the target features. This two-stage approach decouples the complex task of geometry generation from the simpler task of feature prediction. We trained separate models for positive and negative fragments for 100k steps each. In early experiments we found that jointly generating coordinates and features with flow matching resulted in substantially worse MMD scores and that omitting features entirely from the downstream SHAPE classifier reduced its F1 score. We found that predicting curvature was challenging for the regression model; using these noisy predictions in the downstream classifier slightly hurt performance compared to omitting them entirely. We retained the curvature feature in the baseline experiment using only real examples.

## B CONTROLLABLE GENERATION DETAILS

### B.1 CONDITIONING DETAILS

To enable unconditioned generation, the conditioning vector $\mathbf{c}$ is set to a zero vector 80% of the time during training. To enable conditioning only on a subset of the vector, we also provide a binary mask concatenated to the 11-dimensional $\mathbf{c}$, indicating which dimensions are active. Each dimension is masked independently with a probability uniformly sampled from $[0, 1]$ for each sample, resulting in a 2x11-dimensional conditioning vector.

### B.2 CONDITIONING INTERPOLATION DETAILS

Smooth interpolations are generated by varying the conditioning vector $\mathbf{c}$ while keeping the initial noise sample $\mathbf{x}_0$ constant for the entire sequence. For each animation, we define start and end conditioning vectors, $\mathbf{c}_{\text{start}}$ and $\mathbf{c}_{\text{end}}$, and generate intermediate point clouds by linearly interpolating between them. A binary mask, concatenated to $\mathbf{c}$, indicates to the model which feature dimensions are being actively controlled.

The specific value ranges for the interpolations shown in Figure 5 are:

- **Mean x-coordinate:** Interpolated from -0.75 to 0.75.

- **Spatial Extent:** The three diagonal elements of the covariance matrix are interpolated from 0.02 to 0.12.

- **Branching Complexity:** The MST leaves feature is interpolated from 0 to 256.

- **Rotation:** Two diagonal covariance elements are interpolated in opposite directions, one from 0.0 to 0.5 and the other from 0.5 to 0.0, to control elongation and orientation.

Each row in the figure uses a different, randomly chosen but fixed, initial noise sample to demonstrate that the controllable transformation is robust across different initializations.

### B.3 CLASSIFIER-FREE GUIDANCE ABLATION

To enhance control over the generation process, we can employ classifier-free guidance (Ho & Salimans, 2022). It adjusts the predicted vector field by combining the outputs of a conditional and an unconditional model prediction. The guided vector field $\mathbf{v}'$ is computed as

$$\mathbf{v}'_\theta(\mathbf{x}_t, t, \mathbf{c}) = (1 + w)\mathbf{v}_\theta(\mathbf{x}_t, t, \mathbf{c}) - w\mathbf{v}_\theta(\mathbf{x}_t, t, \emptyset), \tag{5}$$

where $w$ is the guidance scale, $\mathbf{c}$ is the conditioning vector, and $\emptyset$ represents no conditioning. Setting $w = 0$ corresponds to standard conditional generation, $w = -1$ recovers unconditional generation. Higher values of $w$ encourage stronger adherence to the conditioning signal, often at the cost of sample diversity.

We performed a detailed ablation study on the classifier-free guidance scale to analyze the trade-off between faithfulness to the conditioning signal and the diversity of the generated samples. The experiment involved testing six different guidance scales across 20 distinct conditioning settings. These settings were derived by taking five discrete steps along each of the four smooth interpolation paths demonstrated in Figure 5 (translation, scaling, branching, and rotation). For each of these 20 conditions, we generated 16 unique samples by starting from different initial noise vectors, allowing us to measure the variance in the output.

We evaluated the results using two metrics, ranking the performance of the six guidance scales for each of the 20 conditions separately:

- **Faithfulness Rank (MSE):** To measure how well a generated sample adheres to its conditioning vector, we first compute the conditioning features from each of the 16 generated samples. We then calculate the Mean Squared Error (MSE) between these computed features and the target conditioning features. A lower MSE indicates better faithfulness.

- **Diversity Rank:** To measure the variety of outputs for a fixed condition, we compute our 10-dimensional evaluation feature vectors for each of the 16 samples. We then calculate the standard deviation along each feature dimension, normalize these values by the standard deviation of the training set, and average them into a single diversity score. A higher standard deviation indicates greater diversity.

The final ranks shown in Table 2 are the average ranks across all 20 conditioning settings. The results confirm the expected trade-off: a high guidance scale of 1.0 achieves the best faithfulness, but also the least diverse samples.

## C FORMAL DEFINITIONS OF GEOMETRIC AND TOPOLOGICAL FEATURES

Let a point cloud be represented by a set of $N$ points $P = \{\mathbf{p}_1, \ldots, \mathbf{p}_N\}$, where each point $\mathbf{p}_i \in \mathbb{R}^3$. We define the features used for conditioning (control) and evaluation as follows.

### C.1 CONTROL FEATURES

These features are used as the conditioning vector $\mathbf{c}$ to guide the generation process.

**Position & Spread (9 features)** These features describe the location and spatial extent of the point cloud.

- **Position (Center of Mass):** The mean vector $\boldsymbol{\mu} \in \mathbb{R}^3$ is the average of all points:

$$\boldsymbol{\mu} = \frac{1}{N} \sum_{i=1}^{N} \mathbf{p}_i \tag{6}$$

- **Spread (Covariance Matrix):** The covariance matrix $\boldsymbol{\Sigma} \in \mathbb{R}^{3 \times 3}$ describes the variance and covariance of the point coordinates. After mean-centering the point cloud, it is computed as:

$$\boldsymbol{\Sigma} = \frac{1}{N-1} \sum_{i=1}^{N} (\mathbf{p}_i - \boldsymbol{\mu})(\mathbf{p}_i - \boldsymbol{\mu})^T \tag{7}$$

As $\boldsymbol{\Sigma}$ is symmetric, we use its 6 unique elements (3 variance terms on the diagonal and 3 covariance terms in the upper triangle) as features.

**Branching Complexity (1 feature)** This feature serves as a proxy for the topological complexity of the neurite fragment.

- **Number of MST Leaves:** We first construct a complete graph where the vertices are all the points $P$ for evaluation or for conditioning a 256-point subsample obtained via farthest point sampling. The edge weights are the Euclidean distances between points. We then find the Minimum Spanning Tree (MST) of this graph, denoted $T$. The number of leaves, $L(T)$, is the count of vertices in $T$ with a degree of 1:

$$L(T) = |\{\mathbf{p}_i \in P \mid \text{degree}_T(\mathbf{p}_i) = 1\}| \tag{8}$$

**Neurite Type (1 feature)** A categorical feature encoding the biological identity of the neurite.

- **Axon vs. Dendrite Encoding:** A scalar value indicates the neurite type:

$$c_{\text{type}} = \begin{cases} +1 & \text{for axon} \\ -1 & \text{for dendrite} \end{cases} \tag{9}$$

## C.2 EVALUATION FEATURES

These 10 rotation-invariant features are used to compute the Maximum Mean Discrepancy (MMD) score between sets of real and generated samples.

**Global Shape (4 features)** These features capture the overall size, position, and elongation of the point cloud.

- **Distance of Center of Mass to Origin:** The Euclidean norm of the mean vector $\boldsymbol{\mu}$:

$$d_{\text{origin}} = \|\boldsymbol{\mu}\|_2 = \sqrt{\mu_x^2 + \mu_y^2 + \mu_z^2} \tag{10}$$

- **Standard Deviation along Principal Components (3 features):** These describe the extent of the point cloud along its principal axes of variation. They are the square roots of the eigenvalues ($\lambda_1 \geq \lambda_2 \geq \lambda_3$) of the covariance matrix $\boldsymbol{\Sigma}$:

$$\sigma_j = \sqrt{\lambda_j} \quad \text{for } j = 1, 2, 3 \tag{11}$$

**Local Point Density (4 features)** These features measure the quality and regularity of the point cloud surface and extent respectively.

- **Mean and Std. Dev. of Nearest Neighbor Distance:** For each point $\mathbf{p}_i$, we find the distance to its nearest neighbor, $d_{1\text{NN}}(\mathbf{p}_i) = \min_{j \neq i} \|\mathbf{p}_i - \mathbf{p}_j\|_2$. We then compute the mean ($\mu_{1\text{NN}}$) and standard deviation ($\sigma_{1\text{NN}}$) of these distances over all points:

$$\mu_{1\text{NN}} = \frac{1}{N} \sum_{i=1}^{N} d_{1\text{NN}}(\mathbf{p}_i) \tag{12}$$

$$\sigma_{1\text{NN}} = \sqrt{\frac{1}{N} \sum_{i=1}^{N} (d_{1\text{NN}}(\mathbf{p}_i) - \mu_{1\text{NN}})^2} \tag{13}$$

- **Mean and Std. Dev. of Farthest Neighbor Distance:** For each point $\mathbf{p}_i$, we find the distance to its farthest neighbor, $d_{\text{far}}(\mathbf{p}_i) = \max_j \|\mathbf{p}_i - \mathbf{p}_j\|_2$. We then compute the mean ($\mu_{\text{far}}$) and standard deviation ($\sigma_{\text{far}}$) of these distances:

$$\mu_{\text{far}} = \frac{1}{N} \sum_{i=1}^{N} d_{\text{far}}(\mathbf{p}_i) \tag{14}$$

$$\sigma_{\text{far}} = \sqrt{\frac{1}{N} \sum_{i=1}^{N} (d_{\text{far}}(\mathbf{p}_i) - \mu_{\text{far}})^2} \tag{15}$$

**Topology (2 features)** Based on an MST constructed on the full point cloud, these features capture the tree-like structure. Let $T$ be the MST and $w(e)$ be the weight of an edge $e \in T$.

- **Total MST Weight:** The sum of all edge weights in the MST, a proxy for the total path length of the neurite skeleton:

$$W(T) = \sum_{e \in T} w(e) \tag{16}$$

- **Longest MST Edge:** The maximum edge weight in the MST, which is sensitive to disconnected components or large gaps:

$$w_{\max}(T) = \max_{e \in T} w(e) \tag{17}$$

## D  EVALUATION PROTOCOL DETAILS

### D.1  CHOICE OF EVALUATION METRIC

Figure 7 illustrates why standard point cloud generation evaluation metrics like Chamfer Distance (CD) are ill-suited for evaluating neuron morphology. Morphologically plausible fragments that differ only by a small change in branch angle can have an increasingly high CD. Our MMD-based metric on interpretable features is more robust to these kinds of valid topological variations. Furthermore, comparing generated samples to a database of known realistic neuron shapes is intractable due to the combinatorial explosion of possible branching patterns that would need to be included in the database.

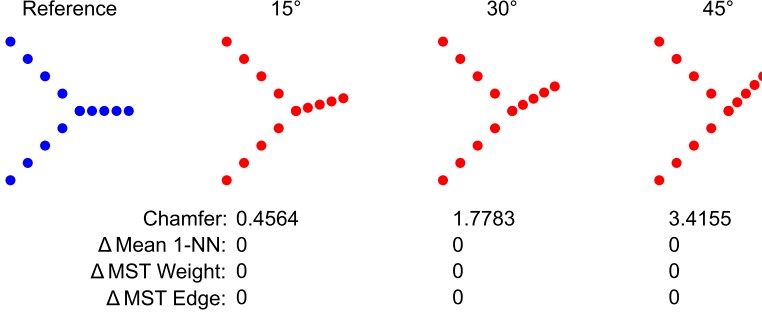

Figure 7: **Why Chamfer Distance is ill suited.** Some of our metrics are invariant to small changes in branching angles while CD increases.

### D.2  MMD DETAILS

To compute the MMD score, we compute 10-dimensional feature vectors for 16,384 real validation and training samples each and 512 generated samples and divide them by the dimension-wise standard deviation of the train set. The MMD is then computed between the normalized embeddings of the real and generated samples. We report MMD scores on the validation set, which closely track performance on the training set.

## D.3   Metric Validation via User Study and UMAP

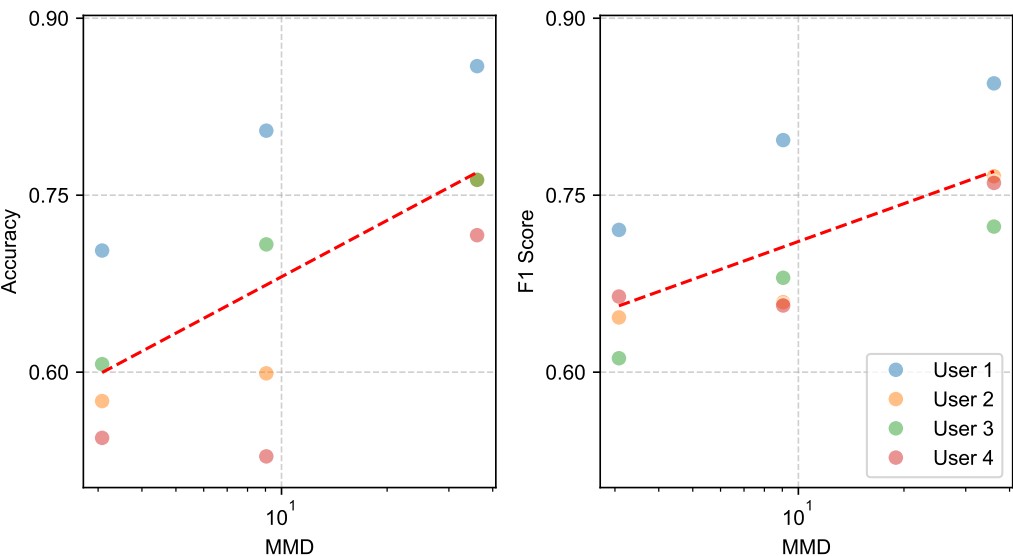

Figure 8: **User study: correlation between MMD and manual real/generated classification performance.** This plot shows the correlation between our quantitative MMD metric (x-axis) and the results of a human evaluation study (y-axis, accuracy/F1 score), demonstrating that lower MMD scores correspond to higher perceived realism. The study involved 4 human evaluators classifying a mix of real samples and 64 generated samples from each of three model variants (ablations with higher MMD scores). To avoid bias, the ratio of real to fake was balanced such that random guessing would yield 50% accuracy. The plot shows a significant correlation between lower MMD (higher model quality) and higher perceived realism.

# E   Downstream Task Details

## E.1   SHAPE Model Training

We trained SHAPE models using 16 A100 GPUs with a batch size of 512, using the AdamW (Loshchilov & Hutter, 2019) optimizer and a learning rate of $5.12 \cdot 10^{-4}$. Training examples were sampled with equal probability from the positive and negative classes, with 10% of the negative examples generated with MoGen. For every checkpoint saved during training (every 1,000 steps) we computed an F1 score using a small subset of the validation set (100 batches of positive and negative examples each). We then took the 20 top scoring checkpoints and performed inference using the full validation set. Unless noted otherwise, we trained and evaluated as described above three differently seeded replicas of every SHAPE model in parallel. We then took the checkpoint with the highest validation F1 score per replica, and used the median checkpoint for the computationally most expensive step of full volume inference and combinatorial search in PATHFINDER (Januszewski et al., 2025).

## E.2   Specialized Generators for Downstream Task

For the PATHFINDER experiments, the final co-training ratio (10%, corresponding to 2.5 M synthetic examples seen during training) and the choice of the optimal model checkpoint were determined via a grid search over a held-out validation set. In early experiments we found that the downstream benefits are maximized when co-training with synthetic examples added to the negative class only. We attribute this to an asymmetry in label noise risk. If the generator creates a Positive sample with an artifact (e.g., a disconnected component), it introduces a problematic False Positive into the training set. Conversely, if the generator creates a Negative (implausible merge) that

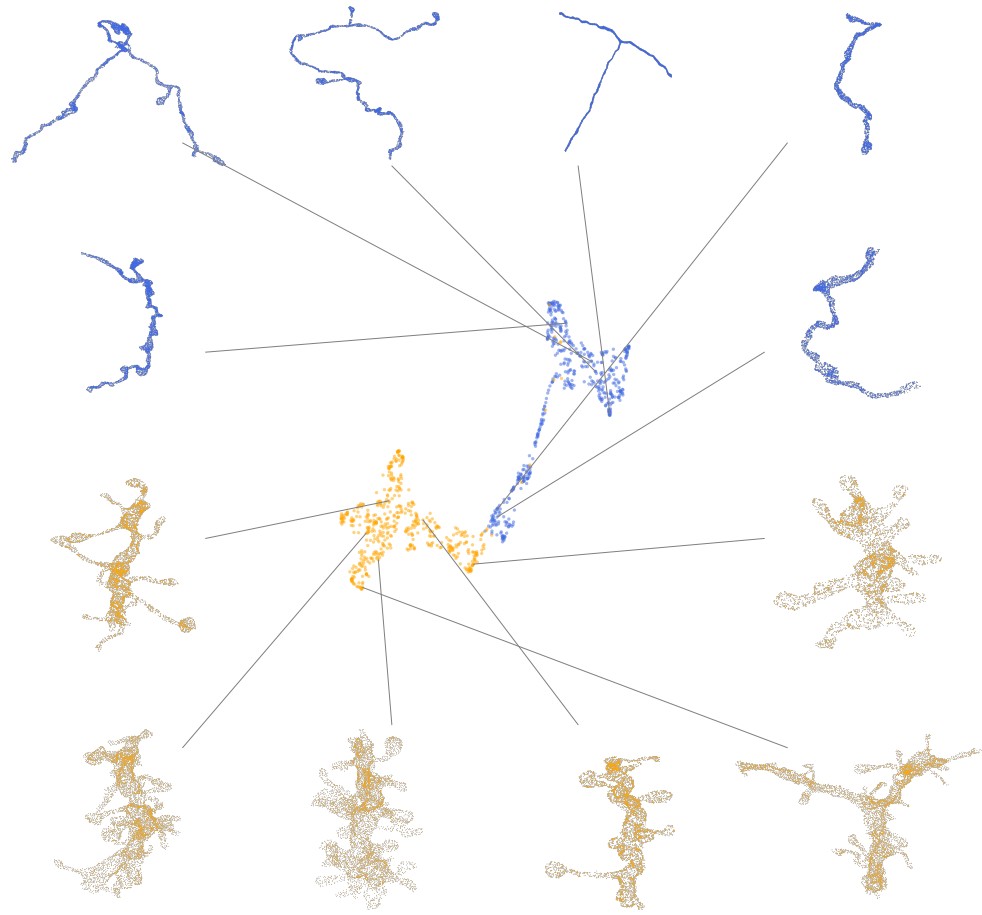

Figure 9: **Metric features validation via UMAP.** A UMAP plot of our 10D feature vectors for real axon (blue) and dendrite (orange) fragments shows clear separation, validating our features' ability to capture meaningful morphological differences. A linear binary classifier achieves 99.3% accuracy on this separation task.

is slightly imperfect, it typically remains a valid negative example. Thus, augmenting the negative class with diverse, hard examples is safer and more effective than augmenting the positive class (see Figure 13).

### E.3  Synthetic Data Replacement Fraction Ablation

### E.4  Manual Labor Calculation

We follow Januszewski et al. (2025) when calculating the expected savings of manual effort needed to eliminate the reconstruction errors via manual proofreading. We take the lower 300,000 person-years workload estimate of Jefferis et al. (2023), reduce it by a factor of 84 due to the use of PATHFINDER (Januszewski et al., 2025), and multiply the result by the 4.4% error reduction shown in our experiments (Figure 6) to account for the additional gains from the use of MoGen-generated examples.

Table 4: Ablation on synthetic data replacement fraction for co-training. We report the highest F1 score achieved by the model in online evaluation (see Appendix E.1).

| Replacement fraction | F1 score (%) ↑ |
|---|---|
| 0% (only real) | 93.91 |
| 1% | 93.93 |
| 5% | 94.10 |
| 10% | **94.64** |
| 50% | 93.65 |
| 90% | 93.43 |
| 100% (only synthetic) | 66.70 |

# F    ADDITIONAL EXPERIMENTS AND QUALITATIVE RESULTS

## F.1    COMPARISON TO LOWER RESOLUTION BASELINE

To situate MoGen within the broader literature on point cloud generation, we performed a comparison against a full transformer baseline inspired by point cloud diffusion approaches (Luo & Hu, 2021) on a down-sampled version of our dataset (2048 points), where this baseline is computationally tractable. As shown in Table 5, MoGen performs competitively while retaining the linear scaling necessary for the full-resolution task. This provides a reference point and highlights the practical necessity of a highly scalable architecture for this scientific application.

The transformer baseline consists of 8 blocks with a 256-dimensional token embedding. Due to the quadratic memory complexity of its self-attention mechanism, deeper models were not feasible at the same batch size and hardware used for MoGen.

Table 5: **Comparison with a baseline on a down-sampled (2048 points) dataset.** MoGen remains competitive in quality while being uniquely suited for scaling to 8192 points. For a fair comparison, the MMD evaluation features were also computed on 2048-point clouds. Both models were trained for 500k steps.

| Model | MMD (Total) ↓ |
|---|---|
| Transformer Baseline | 5.78 |
| **MoGen** | **4.94** |

## F.2    GENERALIZATION TO HIGHER RESOLUTIONS

As noted, our k-NN context injection breaks the formal resolution-invariance of the PointInfinity backbone. While this is an architectural trade-off for achieving high surface quality, we found that in practice the model generalizes also to some degree to higher point cloud resolutions not seen during training. Figure 10 shows examples of high-quality fragments generated with double (works) and quadruple (breaks down), more than the training resolution of 8,192 points, demonstrating practical flexibility despite the theoretical limitation.

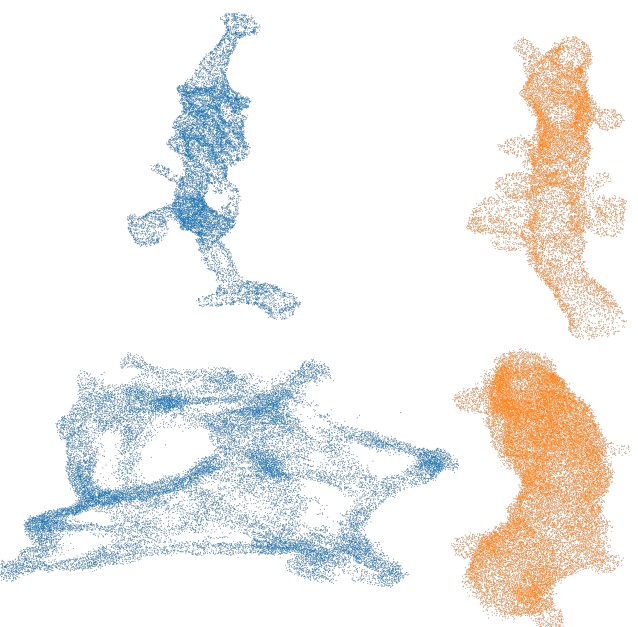

Figure 10: **High-resolution generation.** Example fragments generated at double (top) and quadruple (bottom) the training point count. The model with k-NN context (left) demonstrates an ability to generalize beyond its training resolution, while struggling with larger domain shifts.

## F.3 TYPICAL FAILURE MODES

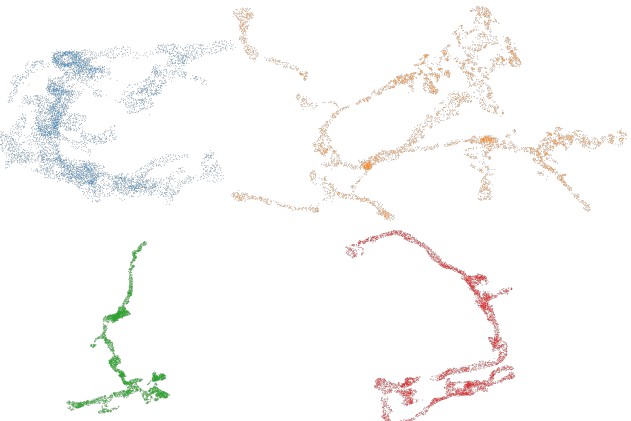

Figure 11: **Failure modes.** Note typical failure modes, such as small disconnected "dust" fragments and unnatural morphologies from a smaller model.

## F.4 QUALITATIVE COMPARISON OF REAL AND GENERATED SAMPLES

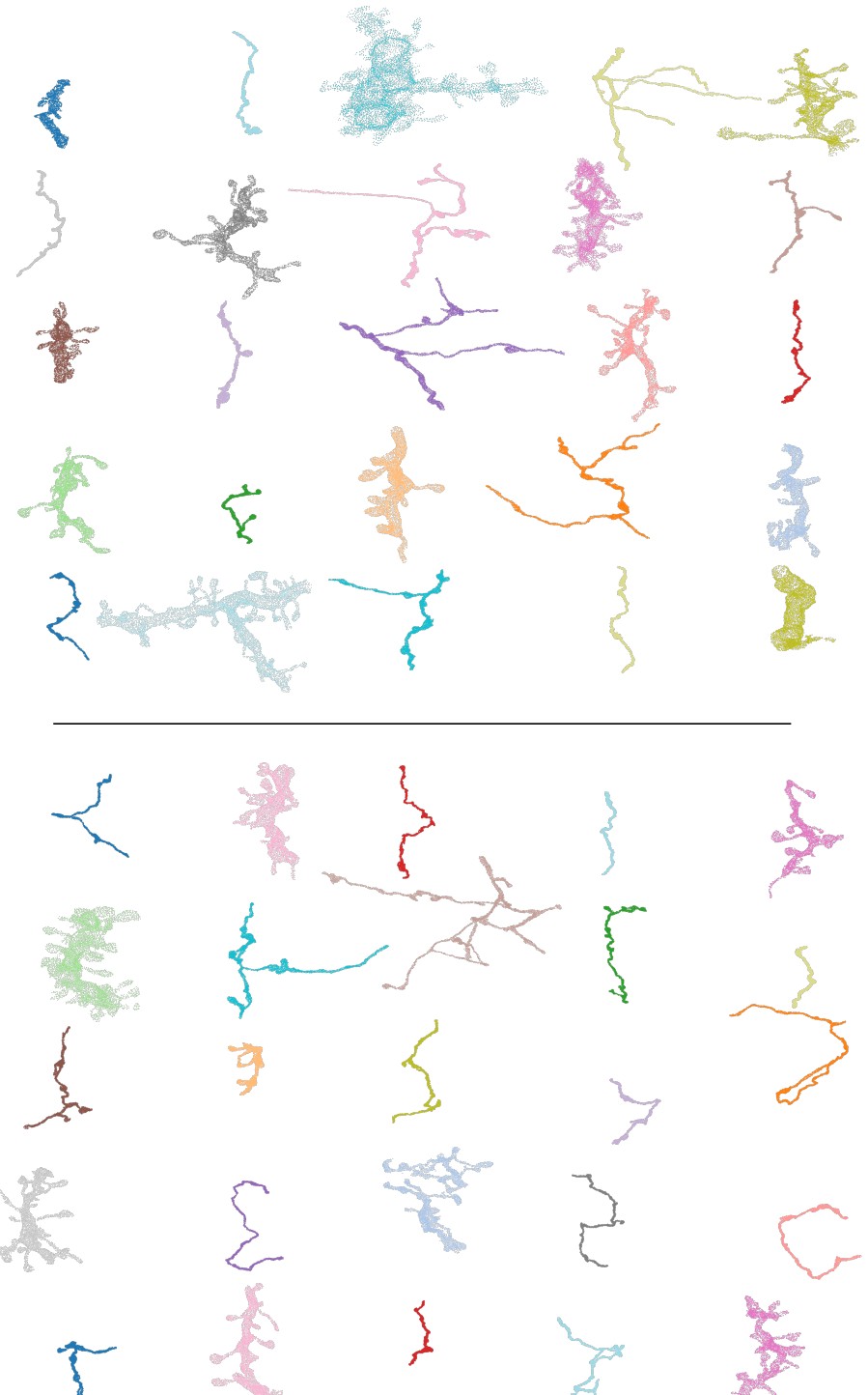

Figure 12: **Real vs. generated mixed samples.** Top: Real samples from the training set. Bottom: Unconditional samples generated by MoGen. The model captures a wide variety of morphologies.

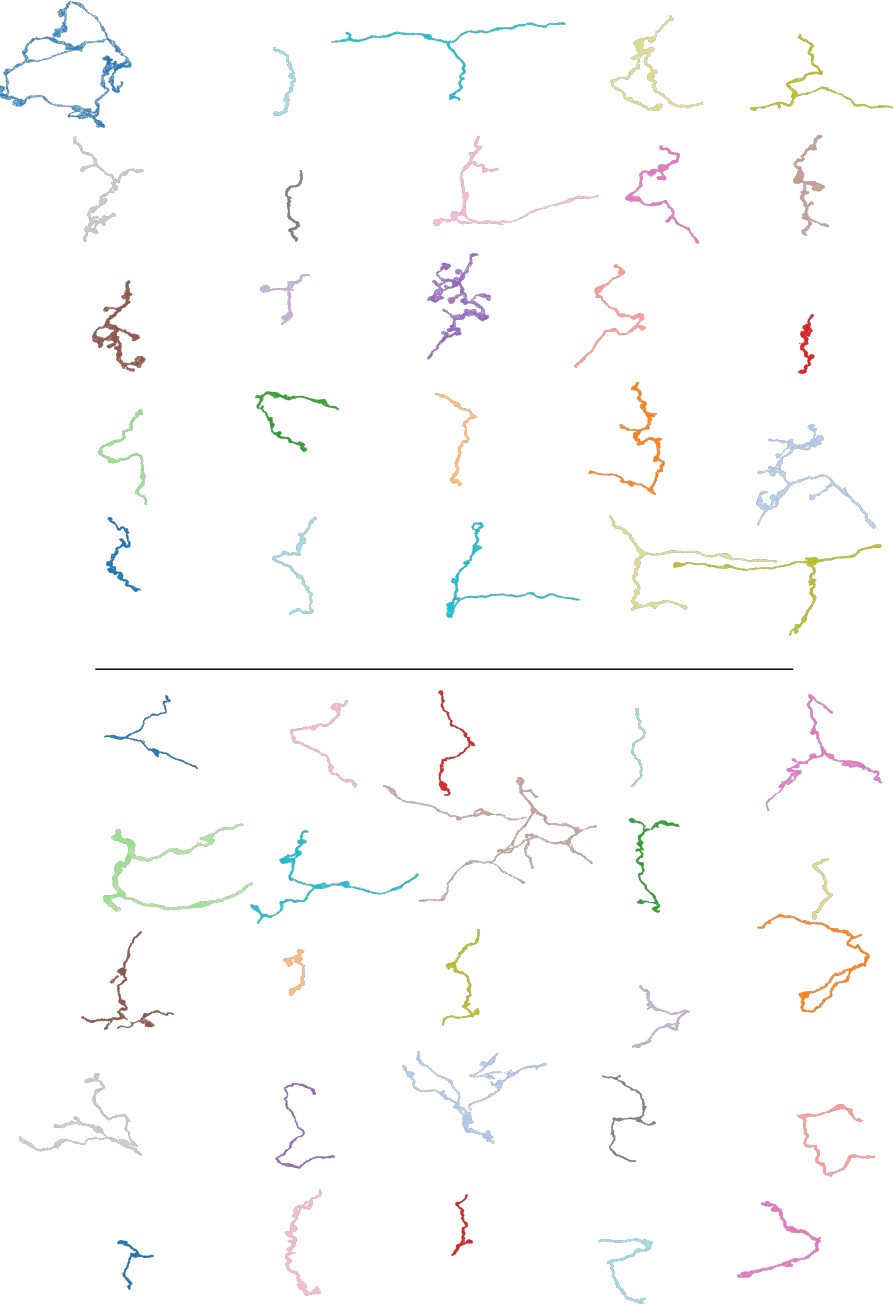

Figure 13: **Real (top) vs. generated (bottom) positive axon samples.** The generated fragments exhibit high morphological fidelity, matching the diversity of real axons.

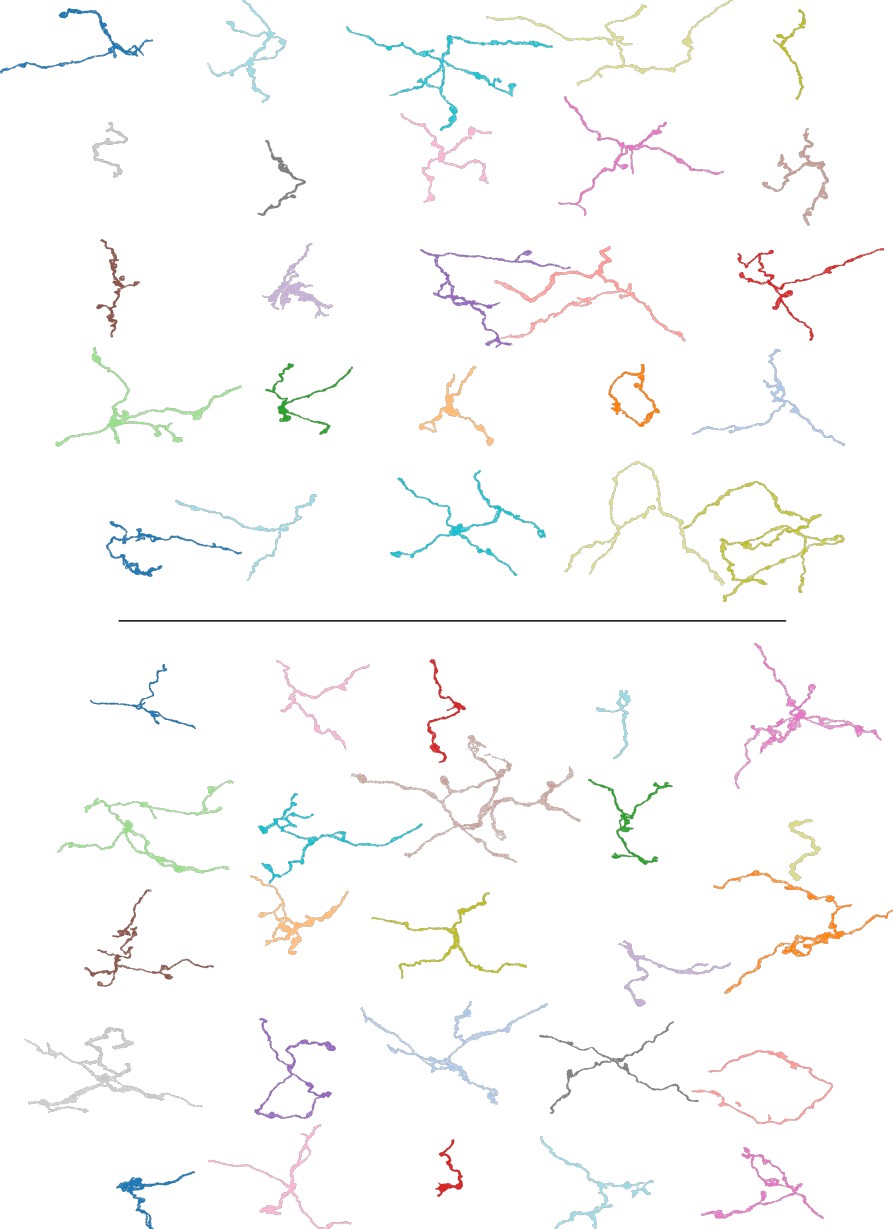

Figure 14: **Real (top) vs. generated (bottom) negative axon samples.** Both real and generated negative samples exhibit specific artifacts such as disconnected components or unnatural mergers, which the SHAPE classifier must learn to reject.

## F.5 Editing Real Samples

Inspired by methods that use generative models for semantic editing (Rout et al., 2025), we demonstrate that MoGen can be used to modify real neuron fragments without any changes to the model architecture. This process involves a "reconstruct and edit" pipeline. First, a real sample $\mathbf{x}_1$ is inverted by integrating the learned ODE backward in time from $t = 1$ to $t = 0$, yielding a noise sample $\mathbf{x}_0$ in the prior distribution. A standard forward integration from $\mathbf{x}_0$ faithfully reconstructs the original sample, $\mathbf{x}'_1 \approx \mathbf{x}_1$. However, by introducing a conditioning vector $\mathbf{c}$ during this forward pass and leveraging classifier-free guidance, we can edit the original sample's properties. As shown in Figure 15, this allows for the creation of counterfactual morphologies, such as increasing the branchiness or altering the spatial extent of a real neuron fragment, providing a powerful tool for exploring morphological variations.

## F.6 Additional Data Experiments

To demonstrate the generalizability of MoGen beyond mouse cortical fragments, we extended our experiments to datasets from different species and acquisition volumes. We prepared point cloud datasets from three additional sources: (1) Drosophila (fruit fly) male central nervous system (Berg et al., 2025) at $10\,\mu\text{m}$ radius, and (2) Zebra finch (bird) basal ganglia (volume j0251) (Rother et al., 2025) at both $10\,\mu\text{m}$ and $50\,\mu\text{m}$ radii. The point cloud extraction followed the same protocol as before (see Section 4.1) sampling 8,192 points within the specified radius centered on random skeleton nodes. Unlike the proofread mouse axon dataset, the zebra finch data was sampled from automated segmentations and includes cell bodies (somata).

We fine-tuned our general MoGen model (pre-trained for 1 million steps on the mixed mouse dataset) on these new datasets. We continued training with the same optimizer state, swapping the data loader to the new target dataset. As shown in Figures 17 and 18, the model adapts well to these domain shifts. It also successfully generates larger-scale structures (Figure 16) and captures distinct morphological features such as somata and the dense branching of medium spiny neurons, which were absent in the original training data.

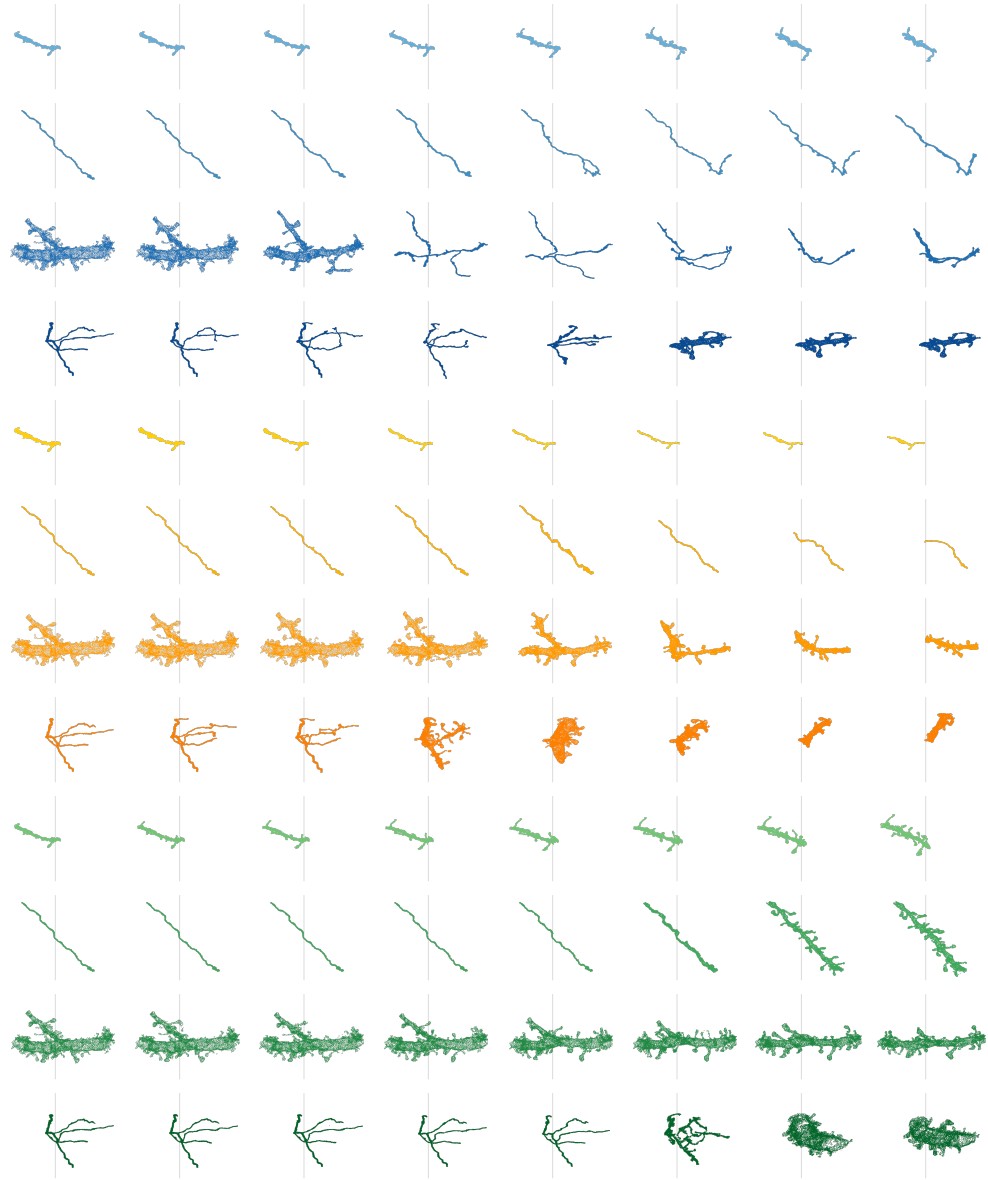

Figure 15: **Editing real samples.** We interpolate the classifier-free guidance scale from $w = -1$ (left, corresponding to the reconstructed real sample) to strong condition adherence at $w = 1$ (right). The rows illustrate editing specific morphological features: Mean x-coordinate (blue) (shifted $+0.2$), Spatial extent (yellow) (reduced to $0.25\times$ original diagonal covariance), and Branching complexity (green) (maximizing MST leaves). Note how the morphology progressively adapts to the target condition as guidance strength increases.

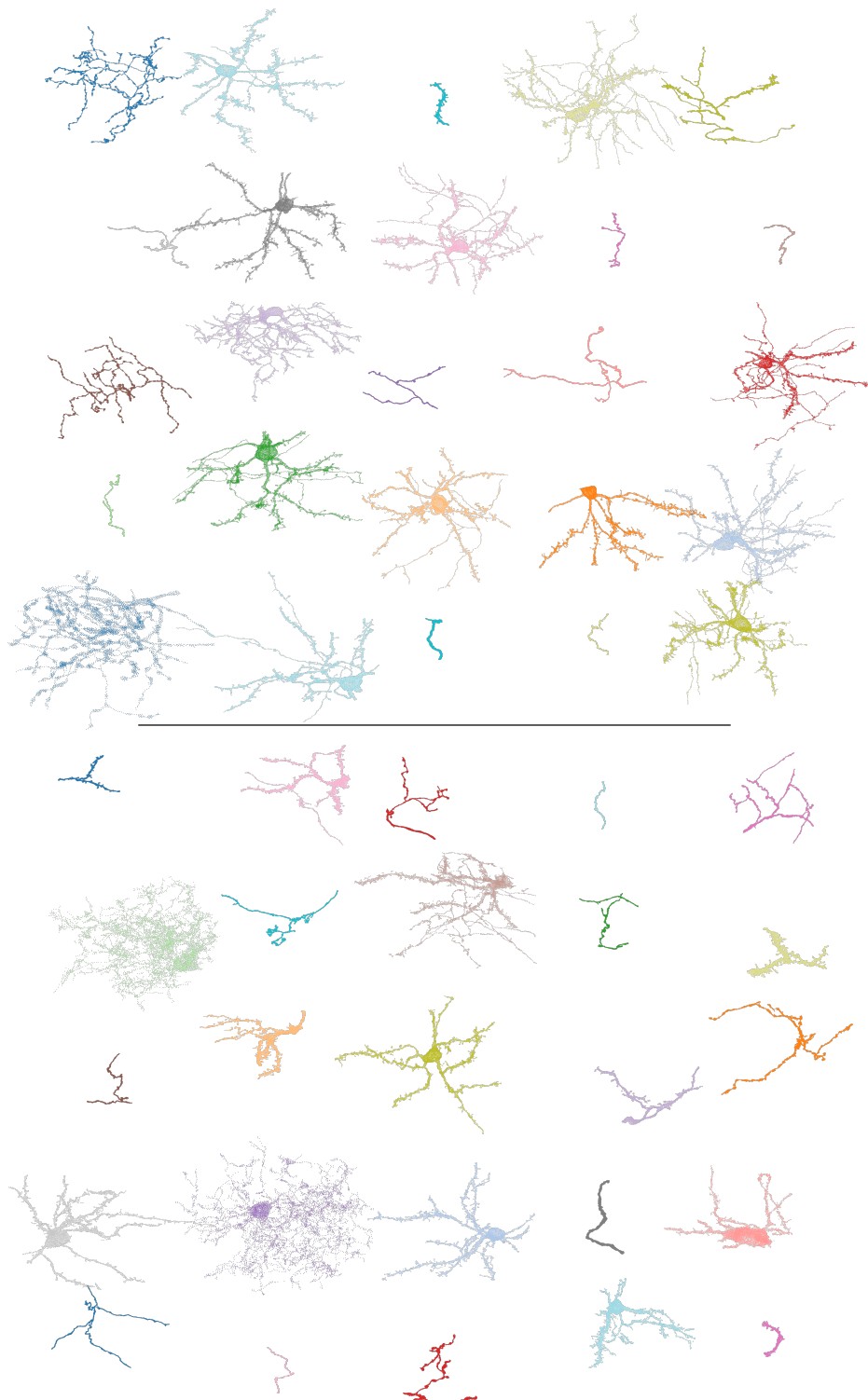

Figure 16: **Generalization to zebra finch (50 microns).** Top: Real samples extracted from the volume. Bottom: Generated samples from the fine-tuned model. Note the successful generation of somata (cell bodies) and characteristic medium spiny neuron morphologies, features not present in the original mouse training set, demonstrating MoGen's capacity to adapt to larger radii and distinct structural classes.

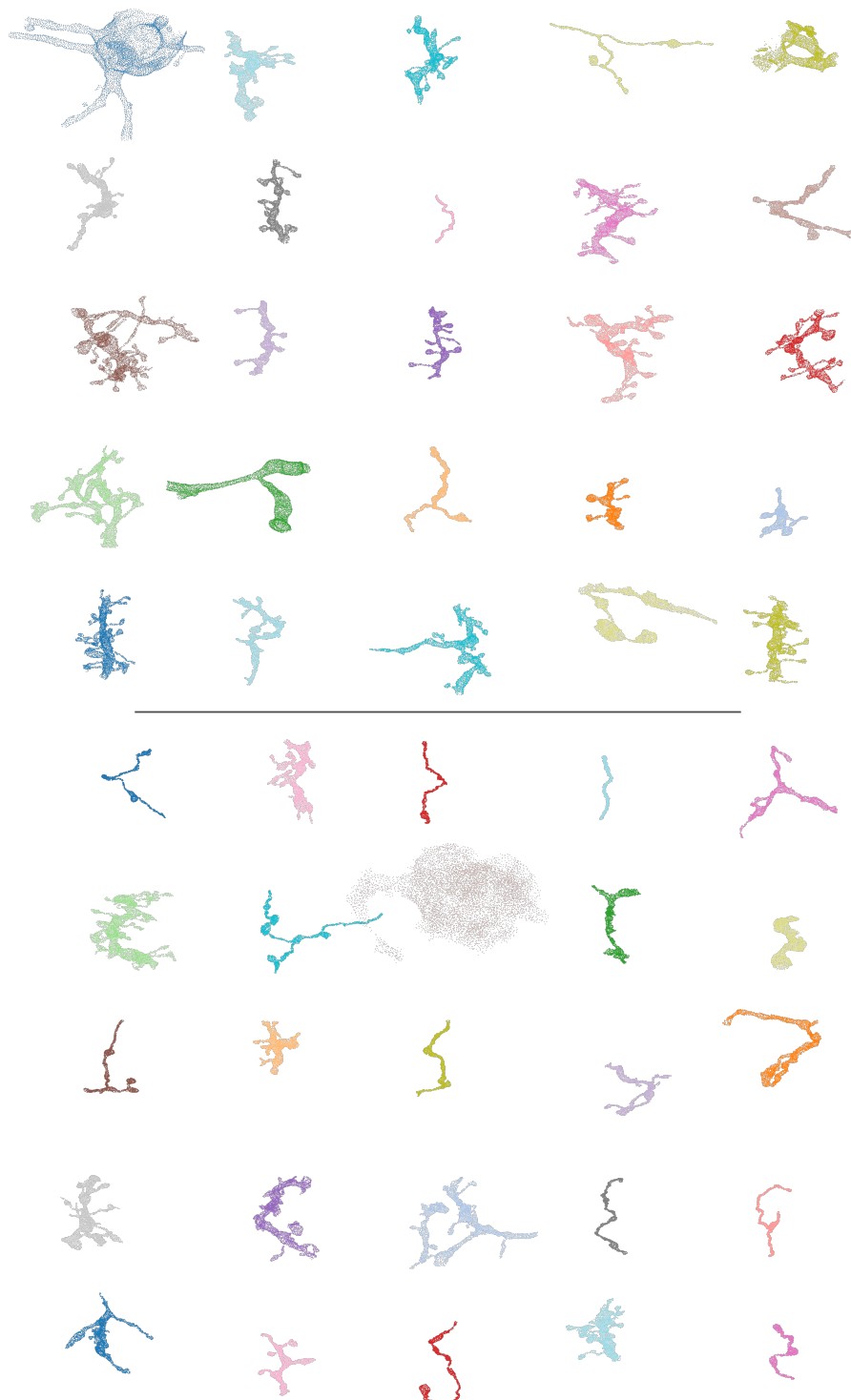

Figure 17: **Generalization to zebra finch (10 microns).** Top: Real samples. Bottom: Generated samples. The model successfully adapts to the specific branching statistics and local geometry of this different species, maintaining high surface fidelity despite the domain shift from the mouse cortex data.

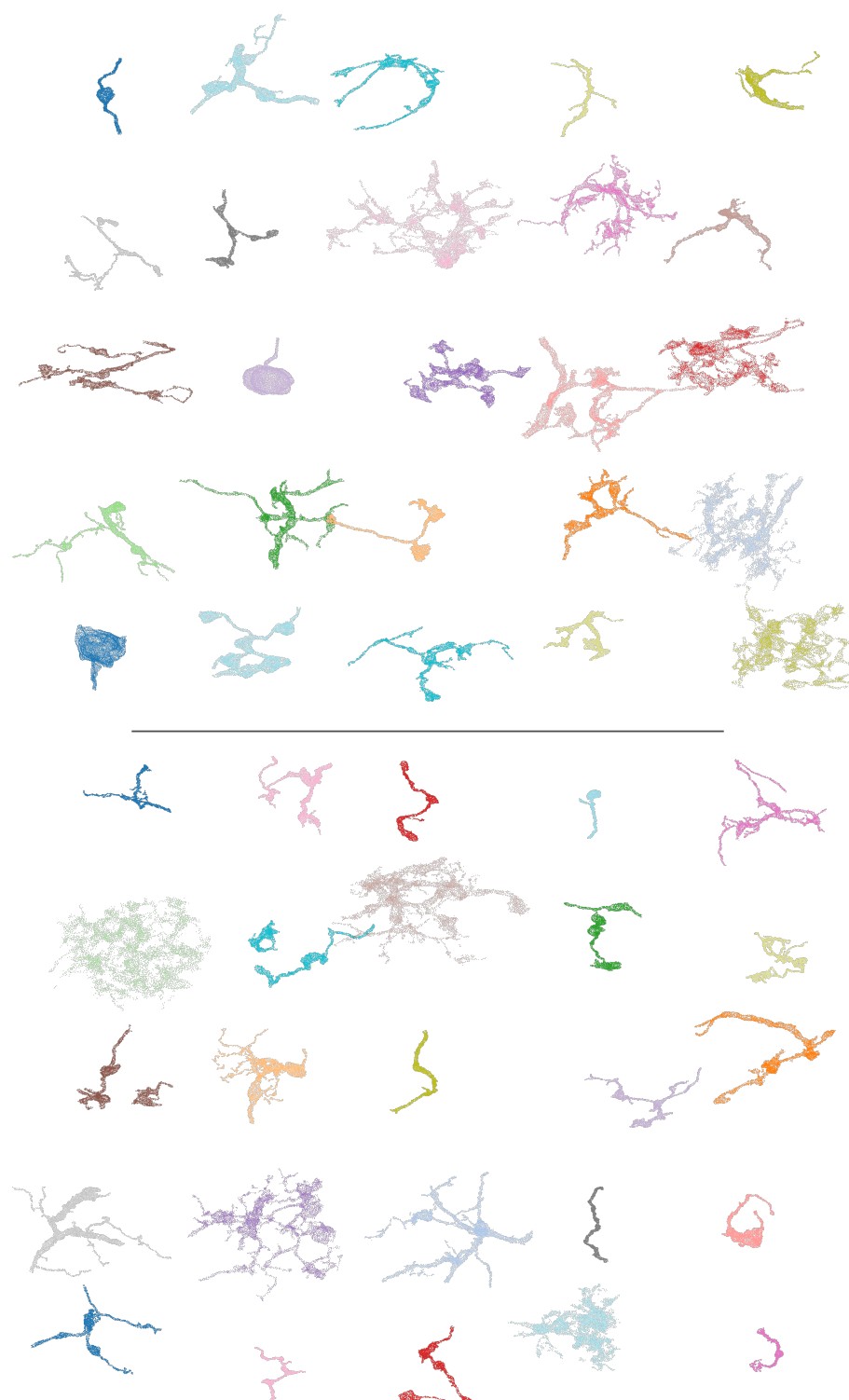

Figure 18: **Generalization to Drosophila (10 microns).** Top: Real samples. Bottom: Generated samples. This demonstrates cross-species generalization to invertebrate neuronal morphologies, capturing the distinct, often more jagged and thin structural properties of Drosophila (fruit fly) neurites.

