# OpenReview forum: "MoGen: Detailed Neuronal Morphology Generation via Point Cloud Flow Matching"
_ICLR.cc/2026/Conference — ICLR 2026 Poster_

### Official Review · Reviewer_99iG · 2025-10-28

**Soundness:** 2
**Presentation:** 3
**Contribution:** 3
**Rating:** 4
**Confidence:** 5

**Summary:**

This paper presents MoGen, a flow matching model for generating high-resolution 3D neuronal morphologies as point clouds. The authors injects local geometric context into a scalable transformer backbone, significantly improving the fidelity of fine details. The authors validate their model with a custom, interpretable evaluation suite and, most importantly, demonstrate its practical utility by augmenting a neural reconstruction pipeline's training data, which leads to a measurable reduction in neuron reconstruction errors.

**Strengths:**

1. The paper employs a state-of-the-art generative architecture, underpinned by a well-motivated modeling insight. The introduction of local geometric context effectively addresses a critical information bottleneck, leading to a demonstrable and significant improvement in the fidelity of generated morphological details.
2. The proposed evaluation suite is a notable contribution in itself. It moves beyond inadequate standard metrics by introducing a set of biologically salient and highly interpretable features that are tailored to the domain-specific properties of neuronal structures, a claim robustly validated through a user study.
3. The work provides a comprehensive and rigorous ablation study that systematically dissects the contribution of each core component. This thoroughly validates the necessity of the proposed architectural adaptations, the model's capacity, and the training schedule.

**Weaknesses:**

1. **The generated data represents a surface/volumetric structure rather than a topological tree.** The outputs of this model are dense point clouds representing surface geometry, which diverges from the classical definition of a neuronal morphology as a binary tree structure (skeleton) [1,2,3,4]. If the authors' goal is to generate detailed cellular surfaces with volume，the work should be more appropriately positioned and compared against prior works that also generate volumetric cell structures, such as [5].
2. **The evaluation protocol for topological features may be compromised by undersampling.** The authors compute two key topological metrics based on a Minimum Spanning Tree (MST) constructed from a heavily subsampled point cloud of only 256 points. This sparse sampling density is likely insufficient to faithfully capture the intricate branching patterns of neurites, potentially leading to the erroneous merging of distinct branches and a significant loss of fine-grained branch morphology details[2]. The validity and sensitivity of these metrics are therefore questionable.
3. **There is no validation of the fundamental biological validity of the generated structures.** A core anatomical property of neurons is that their arborization (excluding the soma) forms a binary tree[6]. The authors do not provide any evaluation to guarantee that the generated point clouds adhere to this essential topological constraint.

**Reference**

[1] Cuntz, Hermann, et al. "One rule to grow them all: a general theory of neuronal branching and its practical application." *PLoS computational biology* 6.8 (2010): e1000877.

[2] Laturnus, Sophie C., and Philipp Berens. "MorphVAE: Generating Neural Morphologies from 3D-Walks using a Variational Autoencoder with Spherical Latent Space." *International Conference on Machine Learning*. PMLR, 2021.

[3] Yang, Nianzu, et al. "MorphGrower: A Synchronized Layer-by-layer Growing Approach for Plausible Neuronal Morphology Generation." *International Conference on Machine Learning*. PMLR, 2024.

[4] Zhu, Tianfang, Hongyang Zhou, and Anan Li. "MorphoGen: Efficient Unconditional Generation of Long-Range Projection Neuronal Morphology via a Global-to-Local Framework." *Proceedings of the IEEE/CVF International Conference on Computer Vision*. 2025.

[5] Wiesner, David, et al. "Implicit neural representations for generative modeling of living cell shapes." *International Conference on Medical Image Computing and Computer-Assisted Intervention*. Cham: Springer Nature Switzerland, 2022.

[6] Peng, Hanchuan, et al. "Morphological diversity of single neurons in molecularly defined cell types." *Nature* 598.7879 (2021): 174-181.

**Questions:**

1. The cells generated in this paper seem much smaller than those in other papers like MorphGrower (as shown in Figure 2). Why is there such a big difference in size? Both are based on mouse brain data, but your generated structures look about ten times smaller.
2. The experimental results show that using the generated data as negative samples for SHAPE classifier training leads to performance improvement. However, if the generated 3D point clouds were highly realistic and biologically plausible, introducing them as *negative* examples would be expected to confuse the classifier and cause performance degradation. The positive result suggests that the generated data **is not biologically realistic**. Instead, it provides a useful set of specific, artificial patterns for the classifier to learn.
3. What is the biological justification for using the mean x-coordinate as a conditioning feature? The manuscript indicates an interpolation range from -0.75 to 0.75. Were all neuronal fragments decentralized during pre-processing? The absolute position of a neuron in the brain is tied to its specific brain region and is unlikely to be confined to such an arbitrary, normalized range. Please clarify the pre-processing steps and the neuroanatomical meaning of this feature.

---

> ### Author Response · Authors · 2025-11-26
>
> We thank the reviewer for the rigorous analysis.
>
> **W1: Motivation for Surface vs. Tree.** The ability to generate surfaces that are not constrained to trees is an intentional contribution, as it allows us to model real-world segmentation artifacts. We focus on this specific scale and detail level for the downstream reconstruction task. While full neurons are often represented as skeleton-trees, methods like MorphGrower miss the volumetric details (thickness, local surface shape, axon terminals) needed for reconstruction from microscopy images. For the SHAPE models in our downstream task, this surface is the main signal. We will add the suggested references [1, 4, 5]. Similar to MorphOcc, Reference [5] lacks these details.
>
> **W2: Evaluation Metric Subsampling.** We have corrected Section 4.2 to clarify this point. The topological features for evaluation are computed on the full-resolution point clouds. The UMAP projection (Figure 9) shows a clear separation between axon and dendrite morphologies. The 256-point subsampling applies exclusively to the calculation of the conditioning vector input to maintain computational efficiency. Our 10 µm radius fragments contain at most dozens of dendritic spines (typical leaf nodes), which are well-preserved by 256 points. This is empirically supported by the smooth morphological changes observed in our interpolation experiments.
>
> **W3: Topological Guarantees.** Real neurite meshes extracted from EM volumes often deviate from perfect trees due to biological self-touches (autapses) or imaging/segmentation artifacts (e.g., disconnected components). We deliberately prioritized **surface fidelity** over topological guarantees, as its high-frequency detail is critical for the downstream task of resolving segmentation errors. Because our model learns the distribution of this real-world surface data, enforcing a strict tree constraint would not always be faithful to the training data and remains as an open research problem for point clouds. Despite the lack of explicit constraints, our evaluations empirically show the model learns to respect biological structure. The User Study (Figure 8) confirms that experts perceive the generated samples as realistic and the downstream application **confirms their usefulness by saving over a hundred years of manual labor**.
>
> **Q1: Cell Size.** The use of neurite fragments instead of full neurons is intentional, as the scale is dictated by the primary downstream task of local error correction with SHAPE models, as we note in our General Response. Modeling whole cells at this level of detail (hundreds of points per micrometer, millions of points overall) would require computational resources beyond the scope of current generative models. Even if we were to reduce the level of detail, we would still be limited by the available datasets, as most current volumes contain many truncated cells whose neurites span beyond the imaged area.
>
> **Q2: Negative Samples & Realism.** It is important to clarify that we trained a specialized MoGen model exclusively on "negative" examples (incorrectly reconstructed fragments) to generate these training samples. These samples are indeed not "biologically realistic" in terms of topology -- they represent segmentation errors. However, they are "plausible-looking" in that they possess the correct local surface statistics of real neurites, but form subtly incorrect geometries. If the generated negatives were unrealistic (e.g., unstructured noise or obviously fake surfaces), the classifier would easily distinguish them, providing no training signal. The fact that co-training improves performance confirms that MoGen generates "hard negatives": synthetic errors that successfully cover the space of plausible but incorrect merges, forcing the classifier to learn more robust decision boundaries. Please also see the General Response.
>
> **Q3: Coordinate Normalization & Conditioning.** The "mean x-coordinate" is a local, relative feature, not an absolute neuroanatomical coordinate. Fragments are extracted by centering a sphere on a skeleton node. This center node becomes the origin (0,0,0) of the point cloud (we have added clarifications about this in Section 4.1). We normalize coordinates so that 1 unit corresponds to the radius of the field of view (10 µm). Therefore, a conditioning value of 0.75 represents a shift of the fragment's center of mass by 7.5 µm relative to the crop center. This feature allows for the manipulation of the fragment's spatial occupancy within the viewing window (as seen in interpolation experiments), rather than placing the neuron in a specific brain region.

---

### Official Review · Reviewer_C4vF · 2025-10-30

**Soundness:** 3
**Presentation:** 3
**Contribution:** 2
**Rating:** 4
**Confidence:** 4

**Summary:**

This manuscript introduces a flow matching generative model for synthesising high-fidelity neurite (axon, dendrite) fragments in the mouse cortex using 3D point clouds. The model adapts the PointInfinity architecture to better model neuronal fragment structures at high resolution, whilst minimizing undesirable discontinuities. This is achieved by injecting local geometric context into the latent cross-attention mechanism that compresses individual point tokens to extract global geometric structures.  The primary practical contribution of this work is to generate additional axon fragment samples to train and improve the performance of a shape plausibility classifier, enabling more reliable merging decisions during neuron reconstructions from EM data.

**Strengths:**

-	The idea of using 3D point clouds for high-fidelity neurite fragments generation is novel and interesting, intuitively suited to the sparse yet expansive volume of the neurites.

-	This is a promising application that could be utilized to improve the final performance of a production-level reconstruction pipeline.

-	The paper is generally well structured and presents the arguments in a logical order. The generation and interpolation visualisations demonstrate the effectiveness of MoGen.

**Weaknesses:**

-	The motivation of this paper is unclear. The paper would benefit from a detailed justification for its narrow focus on neurite fragment generation, neglecting the importance of full neuron topology in characterising neuron identity. From my domain knowledge, in single-neuron classification tasks, the overall structural organization of axons and dendrites often provides more discriminative information than local surface geometry.

-	The value of interpolating from axon to dendrite in the morphological space needs more motivation from a neuroscience perspective. Could the authors explain why this is beneficial?  Biologically, axons would not develop into dendrites and vice versa. Do the smoothly interpolating axon and dendrite morphologies tend to co-exist in real neurons?  Please provide more ablation studies and analysis to address this concern.

-	In the ablation study, the authors validate the effectiveness of classifier-free guidance (cfg). However, flow-matching generative models conventionally use cfg<1.0 to indicate that it doesn’t use the cfg mechanism for sampling. Please explain this further.

-	It is unclear how the axons were categorised into “positive” and “negative” types in Section 4.1. It seems to suggest that negative cases are created by combining real axon morphologies in implausible ways. Additional visualization of these comparisons could help to understand.

-	This manuscript only compares different configurations for the component of their proposed method. How are the comparison results of the existing methods, such as MorphVAE, MorphGrower, and MorphOcc?

-	The downstream application task on improving the SHAPE classifier only supports the usefulness of generated axon fragments and neglects the arguably more intricate dendritic morphologies. Since assessing the model’s impact on a downstream system is another way to validate the generation quality of the neurite fragments, excluding the synthetic dendrites from such evaluation leads to a lack of more rigorous validation for its generation quality, beyond the custom metric via the MMD. The authors may provide further justification for this potential problem.

**Questions:**

•	Why is MoGen only used to generate negative axon samples for the downstream SHAPE co-training task? This seems to suggest that there are enough positive axon examples for training. In this case, they should support a sufficient number of implausible combinations to produce enough negative samples, making the usage of MoGen seem redundant here?

•	Following from the above question, Page 18 states, “In early experiments we found that the downstream benefits are maximized when co-training with synthetic examples added to the negative class only”. Why is this the case? Does this not suggest a lack of fidelity in neuron fragments generated by MoGen?

•	The justification for the 10% negative sample ratio in the downstream task is not clearly expressed. Specifically, what does it mean by “this class of erroneous merges”? Are there different classes of negative examples? If so, why do 10% of the negative examples originate only from one class?

•	Please explain how you define the classifier-free guidance mechanism. Why are there <1.0 values on the cfg?

---

> ### Author Response · Authors · 2025-11-26
>
> We appreciate the reviewer's detailed feedback.
>
> **W1: Motivation (Fragments vs. Whole Neurons).** Please see the General Response. Note that local morphology is also relevant for neuron classification, especially since most datasets only cover brain fragments and thus contain many neurons truncated at the volume boundaries [1].
>
> **W2: Interpolation Motivation.** The interpolation experiment is motivated by three key factors: validating the conditioning mechanism, addressing biological continuity, and enabling counterfactual analysis. This experiment explores the learned branchiness conditioning. It demonstrates the ability of the model to smoothly traverse the complex morphological space between "smooth and thin" axonal features and "spiny and thick" dendritic features. This confirms that the model is not simply retrieving memorized examples but has learned a structured latent representation where high-frequency details (like dendritic spines) can be manipulated via low-dimensional vectors. This is biologically relevant as brains contain axon-dendrite hybrids [2]. The experiment further demonstrates that MoGen allows users to ask "what if" questions, such as "what would this dendrite look like if it had a higher spine density?". This capability turns the generative model into a tool for hypothesis generation. In future datasets with explicit cell-type labels, one could use MoGen for interpolating between neurites of different cell types. This is potentially useful for neuroscientists to discover biologically relevant morphological features similar to [3] or to build biologically realistic simulations with perturbed circuits.
>
> **W3/Q4: CFG Convention.** We initially used a shifted CFG notation and have adapted it now to follow the convention:
> $$v' = (1+w)v_{\text{cond}} - w v_{\text{uncond}}$$
> Consequently, $w=-1$ now corresponds to unconditional generation in our reported tables, while $w=0$ is conditioned generation (without CFG).
>
> **W4/Q1: Positive vs. Negative.** Please see the General Response.
>
> **W5: Comparisons.** As shown in Figure 2, prior methods operate on skeletons or at a completely different scale and level of detail. They cannot generate point clouds corresponding to detailed surface meshes required for our downstream task. Please also see our General Response.
>
> **W6: Motivation for using only axons for downstream evaluation.** The main challenge in neuron reconstruction is very thin axons, which is what PATHFINDER (including its evaluation) was designed for [4]. Dendrites are much bigger and therefore easier to segment. They are currently not supported in the downstream evaluation pipeline or other automated error correction frameworks [5], as in contrast to axons, there is limited room for further improvement with dendrite reconstruction. We validate the quality of dendrite generation through visualization (Figures 2, 5, 12) and the user study (Figure 8).
>
> **Q2:** Please see General Response.
>
> **Q3: 10% justification.** As detailed in Table 4, a 10% co-training ratio achieves best results. Regarding the phrasing "class of erroneous merges": erroneous mergers and negative/implausible are the same class. We have clarified this in the paper now.
>
> [1] Dorkenwald, Sven, et al. "Multi-layered maps of neuropil with segmentation-guided contrastive learning." Nature Methods (2023).
>
> [2] Goaillard, Jean-Marc, et al. "Diversity of axonal and dendritic contributions..." Frontiers in cellular neuroscience (2020).
>
> [3] Eckstein, Nils, et al. "Neurotransmitter classification..." Cell (2024).
>
> [4] Januszewski, Michał, et al. "Accelerating neuron reconstruction with PATHFINDER." bioRxiv (2025).
>
> [5] Schmidt et al. "RoboEM: automated 3D flight tracing for synaptic-resolution connectomics." Nat. Methods (2024).

---

### Official Review · Reviewer_pA2d · 2025-11-03

**Soundness:** 3
**Presentation:** 3
**Contribution:** 3
**Rating:** 6
**Confidence:** 3

**Summary:**

Paper introduces flow-matching method that allows to generate 3D point cloud neuron morphologies by adapting and augmenting standard flow-matching methods e.g. used in image generation. They show significantly improved generations compared to prior work and demonstrate downstream simplifications in human annotation in neuron reconstruction.

**Strengths:**

- Well motivated problem
- Well written paper
- Thorough methods

**Weaknesses:**

- Rather niche and not of broad relevance to ML community. However, I am unsure whether this is a relevant weakness.

**Questions:**

what are other relevant use cases apart from synthetic data generation for tool improvement?

---

> ### Author Response · Authors · 2025-11-26
>
> We thank the reviewer for finding the paper well-written and the method thorough.
>
> **W1: Relevance to ML Community.** The ICLR Call for Papers explicitly invites “applications to neuroscience”. Furthermore, while the application is specific to neuroscience, the problem is a canonical Geometric Deep Learning challenge involving the modeling of high-resolution, non-canonical 3D structures. The Local Context Injection we apply may transfer to other fine-grained point-cloud domains.
>
> **Q1: Other Use Cases beyond the Downstream Classifier.** Besides the visualization, we have added a Semantic Editing experiment (Appendix F.5): By utilizing inverse integration, we can map real neuron fragments to noise and regenerate them with modified conditioning, such as increasing the branchiness (e.g., making smooth axons develop dendritic spines). This serves as a powerful tool for interpreting embeddings or performing future "style transfer" between cell types, serving as counterfactuals useful for neuroscientists to discover biologically relevant features similar to [1].
>
> [1] Eckstein, Nils, et al. "Neurotransmitter classification..." Cell (2024).

---

> > ### Comment · Reviewer_pA2d · 2025-11-27
> >
> > Thank you for the responses. I retain my score.

---

### Official Review · Reviewer_SXdV · 2025-11-12

**Soundness:** 3
**Presentation:** 3
**Contribution:** 2
**Rating:** 6
**Confidence:** 3

**Summary:**

This paper introduces MoGen, a flow-matching generative model for synthesizing high-fidelity 3D neuronal fragments, including axons and dendrites, represented as point clouds. The method adapts the PointInfinity architecture by injecting local geometric context into the latent cross-attention module, improving structural coherence and reducing discontinuities in generated morphologies. The model is trained on EM-derived fragments and evaluated using a custom set of interpretable geometric and topological metrics tailored to neuronal structures. Its main practical contribution lies in augmenting training data for a shape plausibility classifier within the PATHFINDER connectomics pipeline, which enhances automated neuron reconstruction performance.

**Strengths:**

- The paper is clearly written and well-structured, with informative visualizations that effectively illustrate the concepts and generation results.
- The integration of MoGen into the PATHFINDER connectomics pipeline and the demonstrated reduction in reconstruction errors show practical real-world impact.
- The evaluation suite provides interpretable and domain-relevant validation, and its consistency with expert judgments through a user study supports the reliability of the results.

**Weaknesses:**

- Lack of proper discussion with previous 3D neuron synthesis [1] and point cloud representation [2] methods.
- The neuron generation method MoGen is only limited for training shape plausibility classifier within the PATHFINDER setting. How about its application in broader interest such as boosting segmentation/reconstruction directly?
- Since the dataset originates from a specific mouse cortical volume, it is uncertain about MoGen's generalization ability across different datasets.


[1] Tang, Zihao, et al. "3D conditional adversarial learning for synthesizing microscopic neuron image using skeleton-to-neuron translation." 2020 IEEE 17th International Symposium on Biomedical Imaging (ISBI). IEEE, 2020.

[2] Zhao, Runkai, et al. "PointNeuron: 3D neuron reconstruction via geometry and topology learning of point clouds." Proceedings of the IEEE/CVF Winter Conference on Applications of Computer Vision. 2023.

**Questions:**

Please see weakness.

---

> ### Author Response · Authors · 2025-11-26
>
> We thank the reviewer for the positive assessment and for highlighting the practical impact of our work on the PATHFINDER pipeline.
>
> **W1: Related Work.** A key distinction is that the suggested methods utilize skeletons or images thereof, whereas we use volumetric surface points to resolve fine details, which are vital for neuron reconstruction and simulation [1]. We have expanded Section 2 to discuss these references. Please also see our General Response.
>
> **W2: Direct Applications (Segmentation/Reconstruction).** As discussed in our General Response, we designed MoGen to minimize the domain shift, allowing it to operate on the same point cloud domain SHAPE uses. While applying MoGen more directly to segmentation is a conceivable future step (e.g., in PATHFINDER’s SENSE models), it requires further adaptation which is outside of the scope of the current work. Another possible avenue we discuss in Section 6 would be to generate synthetic segmentations from point clouds of multiple neuron instances in a small volume and generate corresponding microscopy images for co-training segmentation models.
>
> **W3: Generalization.** The geometric principles of neurites, such as tubular structures and branching statistics, are shared across vertebrates but are quite distinct between axons and dendrites. We believe the current fragments of three classes (dendrites and plausible/positive as well as implausible/negative axons) provide a robust basis for MoGen's generalization capacity.
>
> [1] Eyal et al. "Human cortical pyramidal neurons: from spines to spikes via models." Front. Cell. Neurosci. (2018).

---

### Author Response · Authors · 2025-11-26
**General Response (1/2)**

We thank the reviewers for their constructive feedback. We appreciate the recognition of the real-world impact of the application (SXdV, 99iG), clarity of presentation (SXdV, pA2d, C4vF), and the novelty of our method (C4vF).

We revised the manuscript to address specific concerns as follows:
*   **Counterfactual Editing:** We added an experiment showcasing an additional application using inverse integration to map real neuron fragments to noise, regenerating with modified conditioning (e.g., "increase branching") (Appendix F.5).
*   **Evaluation Protocol:** We corrected Section 4.2 to clarify that topology metrics are computed on full-resolution point clouds, not subsamples.
*   **Added Related Work:** We discuss additional references in Section 2.
*   **Technical:** We updated the CFG formula convention, updated Table 2, and added coordinate normalization details (Section 4.1).
*   **Visualizations:** We added examples of the classes of “Positive/Plausible” and "Negative/Implausible" samples (Figure 13 and Figure 14).

---

> ### Author Response · Authors · 2025-11-26
> **General Response (2/2)**
>
> We further address shared concerns below:
>
> **Motivation of Detailed Neuron Fragments over Global Topology**
> The primary bottleneck in connectomics is the correction of automated segmentation errors in large (electron) microscopy volumes. This "proofreading" process, historically done manually by human experts, is estimated to cost billions of USD for a single mouse brain [1]. To automate this, systems such as PATHFINDER [2] use classifiers to decide if a specific merge between two segments (neuron fragments) is valid. In PATHFINDER this is accomplished with a point cloud plausibility classifier ("SHAPE"). SHAPE relies heavily on local surface geometry on the scale of microns, rather than the global shape of the neuron which can span many millimeters. SHAPE is training data-limited. Hence, with MoGen, we focus on generating samples in the same domain as SHAPE.
>
> Much prior work models full neurons' global topology (skeletons, potentially with coarse radii or occlusion maps) as occasionally used in cell type classification. In contrast, we target local neuronal membrane morphology at the level of neuron fragments. This focus is crucial for agglomerating segmentation fragments into full neurons [2]. The downstream SHAPE classifier relies on high-fidelity surface details (caliber changes, axon boutons) which skeletons discard. MoGen is the first generative model to address this high-precision regime and to showcase substantial labor reduction in downstream applications. This difference in objective explains the size difference noted by Reviewer 99iG: we generate the high-resolution "building blocks" of neurons to aid reconstruction, rather than low-resolution whole-cell skeletons.
>
> **"Positive" vs. "Negative" Class & Downstream Application**
> The downstream SHAPE classifier is binary: the “Positive” class comprises plausible, manually proofread, correctly agglomerated neurites, while the “Negative” class consists of implausible, erroneous merges. Early experiments showed that while augmenting the Negative class provided significant benefits, augmenting the Positive class did not. We attribute this to asymmetries in how examples from the two classes are generated and in label noise risk regarding generative artifacts.
>
> While positive examples can be sampled directly from the volumetric dataset of reconstructed axons, identifying relevant negative examples is more difficult as it relies on the predictions of a separate model in the PATHFINDER system called SENSE [2]. SENSE acts as a high-recall proposal generator — it analyzes a segment’s local image context to predict its likely continuation into the surrounding volume, thereby building an overcomplete graph of potential connections. Obtaining a sufficient variety of relevant negative examples is difficult because they must resemble the specific false continuations that SENSE is prone to predicting. To address this, MoGen serves as a targeted hard-negative miner. By generating synthetic examples, MoGen populates the "SENSE-relevant" space with diverse instances of erroneous merges — shapes that look locally convincing enough to be proposed as continuations, but are incorrect. This helps the SHAPE model to remain robust against the broad spectrum of ambiguous structures it will encounter during inference.
>
> No generative model captures the distribution perfectly and as discussed in Section 6, MoGen occasionally produces topological errors, such as disconnected components or loops. If the generator attempts to create a 'Positive' but introduces such an error, it creates a False Positive — effectively teaching the classifier that broken anatomy is plausible. Conversely, an imperfectly generated 'Negative' containing disconnected components or loops remains a valid Negative. Because of this asymmetric label noise risk, the penalty for generation artifacts is negligible for the Negative class but detrimental for the Positive class.
>
> By strictly targeting MoGen to model the diverse and under-sampled distribution of erroneous merges (“Negatives”), we achieve our primary goal of improving classifier accuracy. This approach translates to an estimated **saving of ~157 person-years of manual labor** for a whole mouse brain project.
>
> **References:**
>
> [1] Jefferis, Gregory, et al. "Scaling up connectomics: The road to a whole mouse brain connectome." Wellcome (2023).
>
> [2] Januszewski, Michał, et al. "Accelerating neuron reconstruction with PATHFINDER." bioRxiv (2025).

---

> > ### Author Response · Authors · 2025-12-03
> >
> > Dear Area Chair,
> >
> > In light of the recent AC reassignment, we wanted to highlight a final revision of our paper that we have just uploaded, which includes a new section, **Appendix F.6** (Additional Data Experiments), along with Figures 16, 17, and 18.
> >
> > This section provides evidence addressing a specific concern raised by Reviewer SXdV regarding MoGen's ability to generalize across different datasets. We demonstrate that **MoGen successfully adapts to neuroscience datasets** from different species, drosophila (fruit fly) and zebra finch, and different morphological scales (including somata and larger radii) via fine-tuning.
> >
> > We hope this additional evidence aids you in your assessment.
> >
> > Sincerely,
> >
> > The Authors

---

### Meta-Review · Area_Chair_5XWP · 2026-01-13

**Summary:**

The paper introduces a new flow-matching based generative model for high-resolution 3d point clouds representing fragments of neurites (axons and dendrites). The authors use this generative model to improve a shape plausibility classifier and show a modest improvement in accuracy.

The paper received mixed reviews with common concerns on motivation and scope. However, the authors provided a strong rebuttal which would likely have addressed the concerns of the negative reviewers, leading to a weak accept.

**Reviewer Concerns:**

Major concerns included motivation for the need for a neuron fragment level generative model and use cases and generalization across data from different species. The rebuttal successfully addressed these concerns with clarifications and additional experiments.

**Reviewer Scores:**

I suspect the positive reviewers would likely keep their weak accept scores of (6, 6) and the negative reviewers would also have raised their scores to a weak accept (4 -> 6, 4 -> 6).

---

### Decision · Program_Chairs · 2026-01-26

Accept (Poster)